# Positive regulation of oxidative phosphorylation by nuclear myosin 1 protects cells from metabolic reprogramming and tumorigenesis in mice

Tomas Venit[1], Oscar Sapkota[1], Wael Said Abdrabou [1,2], Palanikumar Loganathan [1], Renu Pasricha[3], Syed Raza Mahmood [2], Nadine Hosny El Said [1], Shimaa Sherif [4], Sneha Thomas[3], Salah Abdelrazig[1], Shady Amin [1], Davide Bedognetti [4,5,6], Youssef Idaghdour [1,2], Mazin Magzoub [1] & Piergiorgio Percipalle [1,2,7] ✉

Metabolic reprogramming is one of the hallmarks of tumorigenesis. Here, we show that nuclear myosin 1 (NM1) serves as a key regulator of cellular metabolism. NM1 directly affects mitochondrial oxidative phosphorylation (OXPHOS) by regulating mitochondrial transcription factors TFAM and PGC1α, and its deletion leads to underdeveloped mitochondria inner cristae and mitochondrial redistribution within the cell. These changes are associated with reduced OXPHOS gene expression, decreased mitochondrial DNA copy number, and deregulated mitochondrial dynamics, which lead to metabolic reprogramming of NM1 KO cells from OXPHOS to aerobic glycolysis.This, in turn, is associated with a metabolomic profile typical for cancer cells, namely increased amino acid-, fatty acid-, and sugar metabolism, and increased glucose uptake, lactate production, and intracellular acidity. NM1 KO cells form solid tumors in a mouse model, suggesting that the metabolic switch towards aerobic glycolysis provides a sufficient carcinogenic signal. We suggest that NM1 plays a role as a tumor suppressor and that NM1 depletion may contribute to the Warburg effect at the onset of tumorigenesis.

Functional mitochondria are crucial for a healthy cell as they maintain intracellular calcium levels, communicate with the nucleus via metabolites produced by the Krebs cycle to initiate epigenetic changes and modulate their dynamics to fit the bio-energetic demands of cells[1–4]. However, their primary role is to produce energy in the form of up to 36 ATP molecules via OXPHOS. In hypoxic conditions, cells switch to the less efficient glycolysis pathway, which converts glucose to lactate and produces only 2 molecules of ATP per molecule of glucose. As the majority of cells use OXPHOS as a primary energy source, the expression of both nuclear and mitochondrial genes encoding macromolecular complexes involved in the OXPHOS electron transport chain is tightly regulated[5]. This is not true for highly proliferating

[1]Program in Biology, Division of Science and Mathematics, New York University Abu Dhabi (NYUAD), P.O. Box, 129188 Abu Dhabi, United Arab Emirates. [2]Center for Genomics and Systems Biology, New York University Abu Dhabi (NYUAD), P.O. Box, 129188 Abu Dhabi, United Arab Emirates. [3]Core Technology Platforms, New York University Abu Dhabi (NYUAD), P.O. Box, 129188 Abu Dhabi, United Arab Emirates. [4]Translational Medicine Department, Research Branch, Sidra Medicine, Doha, Qatar. [5]Department of Internal Medicine and Medical Specialties (DiMI), University of Genoa, Genoa, Italy. [6]College of Health and Life Sciences, Hamad Bin Khalifa University, Qatar Foundation, Doha, Qatar. [7]Department of Molecular Biosciences, The Wenner-Gren Institute, Stockholm University, SE-106 91 Stockholm, Sweden. ✉e-mail: pp69@nyu.edu

undifferentiated pluripotent and cancer stem cells, which use glycolysis as a primary source of energy production even in the presence of oxygen. The so-called aerobic glycolysis or Warburg effect was initially explained as a consequence of dysfunctional mitochondria in cancer cells. Nowadays it is highly accepted that aerobic glycolysis in these cells does not serve as a rescue mechanism for defective mitochondria but it is a rather universal, highly regulated metabolic pathway, which, even though less energetically effective, provides advantages to the cells. Glycolysis produces ATP faster than OXPHOS, as a process, it is less dependent on environmental factors, it regulates the tumor microenvironment by increasing intra- and extracellular acidity, and it allows signal transduction through different secondary messengers and promotes flux of byproducts into biosynthetic pathways[6]. Metabolic switches characterized by increased mitochondrial OXPHOS and decreased glycolysis to produce ATP are key features that mark the differentiation of progenitor cells to committed cell lineages[7]. Similarly, metabolic switches are observed in tumorigenesis, where the prevalence of glycolytic metabolism over OXPHOS is connected to poor survival prognosis in many different types of cancers[8]. Mitochondrial metabolism and more specifically, the relationship between OXPHOS and aerobic glycolysis plays, therefore, an important role in key cellular processes such as stemness and differentiation, but is also a defining feature during carcinogenesis. Several cytoskeletal proteins such β actin, Myosin II, or Myosin XIX have been shown to regulate mitochondrial dynamics directly[9–12] with some of them even present within the mitochondria[13,14]. However, changes in metabolism from OXPHOS to glycolysis likely require complex gene expression regulation, which is yet to be fully understood.

Nuclear myosin 1 (NM1), a Myosin 1C isoform, facilitates transcription activation as part of the chromatin remodeling complex B-WICH, with the ATPase subunit SNF2h and WSTF[15–19]. NM1 interacts with SNF2h, enabling nucleosome repositioning and NM1-dependent recruitment of the histone acetyl-transferase (HAT) PCAF and the histone methyl-transferase (HMT) Set1B, to maintain and preserve H3K9-acetylation (H3K9Ac) and H3K4-trimethylation (H3K4me3). This, in turn, leads to a chromatin landscape favorable for transcription activation and elongation[19–21], suggesting NM1 as a global transcriptional regulator[22,23]. Several studies have even proposed that products of the *Myo1C* gene, including NM1, may be possible tumor suppressors as the gene itself is often mutated in various types of cancers[24,25]. In addition to this, we have recently shown that NM1 deletion leads to genome instability, elevated cell proliferation, and increased DNA damage[19,23]. As these are all typical hallmarks of cancer cells, we examined whether dysregulated transcription upon NM1 depletion correlates with changes in the metabolic pathway these cells use to tackle their energetic needs.

Here we show that metabolic reprogramming is transcriptionally regulated by NM1 and this has a direct effect on mitochondrial biogenesis and function. Indeed, NM1 loss leads to a metabolic switch from OXPHOS to aerobic glycolysis and therewith associated mitochondrial phenotypes, such as reduction of mitochondrial networks, underdeveloped inner membrane cristae, and transcriptomic changes in mitochondrial biogenesis. NM1 knockout (KO) cells exhibited transcriptional and metabolome profiles typical of cancer cells and were found to form solid tumors in mice. Mechanistically, we found that NM1 activates the expression of specific mitochondrial transcription factors working as effectors of the PI3K/AKT/mTOR axis and NM1 is part of a positive feedback loop with mTOR. We suggest a pathway in which, upon stimulation, mTORC1 regulates NM1 which then binds to the transcription start site (TSS) of mitochondrial transcription factors PGC1α and TFAM as well as to mTOR itself. This leads to an accumulation of mitochondrial factors, increased mitochondrial biogenesis, and expression of additional mTOR protein that can further activate NM1 until stimuli persist. We speculate that these mechanisms are important during tumorigenesis and that a positive feedback loop

between mTOR and NM1 can play a role in cancer cell survival during mTOR-targeted cancer therapies.

## Results

### NM1 deletion suppresses the expression of OXPHOS genes and dysregulates the expression of mitochondrial genes

Seeing the role of NM1 in the transcriptional response to DNA damage[23], we specifically investigated the potential effect of NM1 deletion on the expression of mitochondrial genes. Total protein extracts from stable NM1 WT mouse embryonic fibroblasts (MEFs), NM1 KO MEFs, and NM1 KO MEFs expressing exogenous NM1 (KO + NM1) were subjected to immunoblotting with the Total OXPHOS Rodent WB Antibody Cocktail kit containing antibodies against subunits of the five complexes (complexes I – V) in the OXPHOS system - Nduf88, Sdhb, Uqcrc2, MtCo1, and ATP5a (Fig. 1a). Results from these experiments show that NM1 deletion leads to a decrease in the amount of each of the protein components of OXPHOS complexes in comparison to the WT condition, while NM1 reintroduction in the KO background fully rescues the OXPHOS protein levels. Interestingly, the higher NM1 expression level observed in the knock-in cells (KO + NM1) in comparison to WT seems to be associated with higher expression of OXPHOS proteins suggesting that OXPHOS protein expression levels are dependent on the amount of NM1 protein in cells (Fig. 1a). RTqPCR analysis of mitochondrial-encoded OXPHOS genes (mt-CO1, mt-Cyb, and mt-ND1) and nuclear-encoded OXPHOS genes, representing all five subunits of the OXPHOS chain (Ndufs1, SDHA, UQCRB, Cox5A, and ATPf51) confirmed the results from immunoblots (Fig. 1b, c). Results from RTqPCR analysis show that expression of all OXPHOS genes is reduced in KO cells while NM1 reintroduction in the KO background (KO + NM1) shows increased expression of each gene even in comparison to WT cells, proving the specificity of the NM1 KO system (Fig. 1b, c). Additionally, we analyzed previously published RNA sequencing data from primary mouse embryonic fibroblasts derived from NM1 WT and KO embryos[23]. NM1 KO mice were prepared by homologous recombination of LoxP sites into the NM1 gene and subsequent Cre-dependent excision of the NM1 start codon. As this system is not dependent on CRISPR/Cas9 technology, the possibility of off-target effects is minimal and it can therefore serve as an additional control to check the specificity of the observed phenotypes in stable NM1 KO cells[26]. We found a substantial portion of OXPHOS genes to be differentially expressed in NM1 KO primary fibroblasts and even though some genes such as Uqcrb show opposite expression patterns in comparison to stable KO cells, the majority of differentially expressed genes are suppressed in primary NM1 KO cells corroborating the above findings and supporting previous results observed in stable NM1 KO MEFs (Fig. 1d). The comparison of all differentially expressed genes with all genes under the MGI Gene Ontology (GO) term "Mitochondrion" (GO:0005739) revealed a high correlation between the groups with over 40% of all mitochondria-associated genes being differentially expressed in the NM1 KO condition (Fig. 1e) affecting processes such as mitochondrial organization, translation, and transport together with previously described OXPHOS (Fig. 1f). As nuclear- and mitochondrial-encoded OXPHOS genes are suppressed and mitochondrial genes are overall deregulated upon NM1 depletion, we next measured mitochondrial DNA copy number which serves as a biomarker of mitochondrial function[27]. We performed quantitative-PCR (qPCR) of genomic DNA (nuclear and mitochondrial) isolated from WT, KO, and KO + NM1 MEFs and compared the relative abundances of three mitochondrial genes mt-Cyb, mt-Nd1, and mt-ATP6 normalized to the nuclear-encoded Terf gene[14]. Results from these experiments show that the lack of NM1 correlates with a reduction of mitochondrial copy number and is fully restored upon reintroduction of NM1 (Fig. 1g). Taken together, these results indicate a wide transcriptional change in mitochondria-related gene

expression accompanied by downregulation of OXPHOS genes and reduction of mitochondrial copy number in NM1 deficient cells.

## NM1 depletion leads to reduced mitochondrial mass, perinuclear localization, and altered mitochondrial morphology and structure

To address the effect of NM1 deletion on mitochondrial structure and function, we first quantified the mitochondrial mass by spectrophotometric analysis of MitoTracker DR staining normalized to nuclear Hoechst staining. Following previous results, NM1 KO cells showed decreased mitochondrial staining in comparison to WT cells with full rescue in the KO + NM1 cells (Fig. 2a). Mitochondria staining

with mitochondrial membrane potential-dependent MitoTracker Orange dye showed a similar decrease in NM1 KO cells compatible with decreased total mitochondrial mass and, remarkably, NM1 reintroduction (KO + NM1 cells) led to a partial rescue of mitochondrial membrane potential (Fig. 2b). To address this further, we used high-content phenotypic profiling to measure mitochondrial mass and mitochondrial distribution across the cells by using MitoTracker DeepRed (MitoTracker DR) that covalently binds thiol groups of cysteine residues of mitochondrial proteins, providing a consistent signal along mitochondria[28]. For data analysis, we used the Compartment Analysis BioApplication software that generates masks to measure quantitative parameters. The mask covering the whole cell except

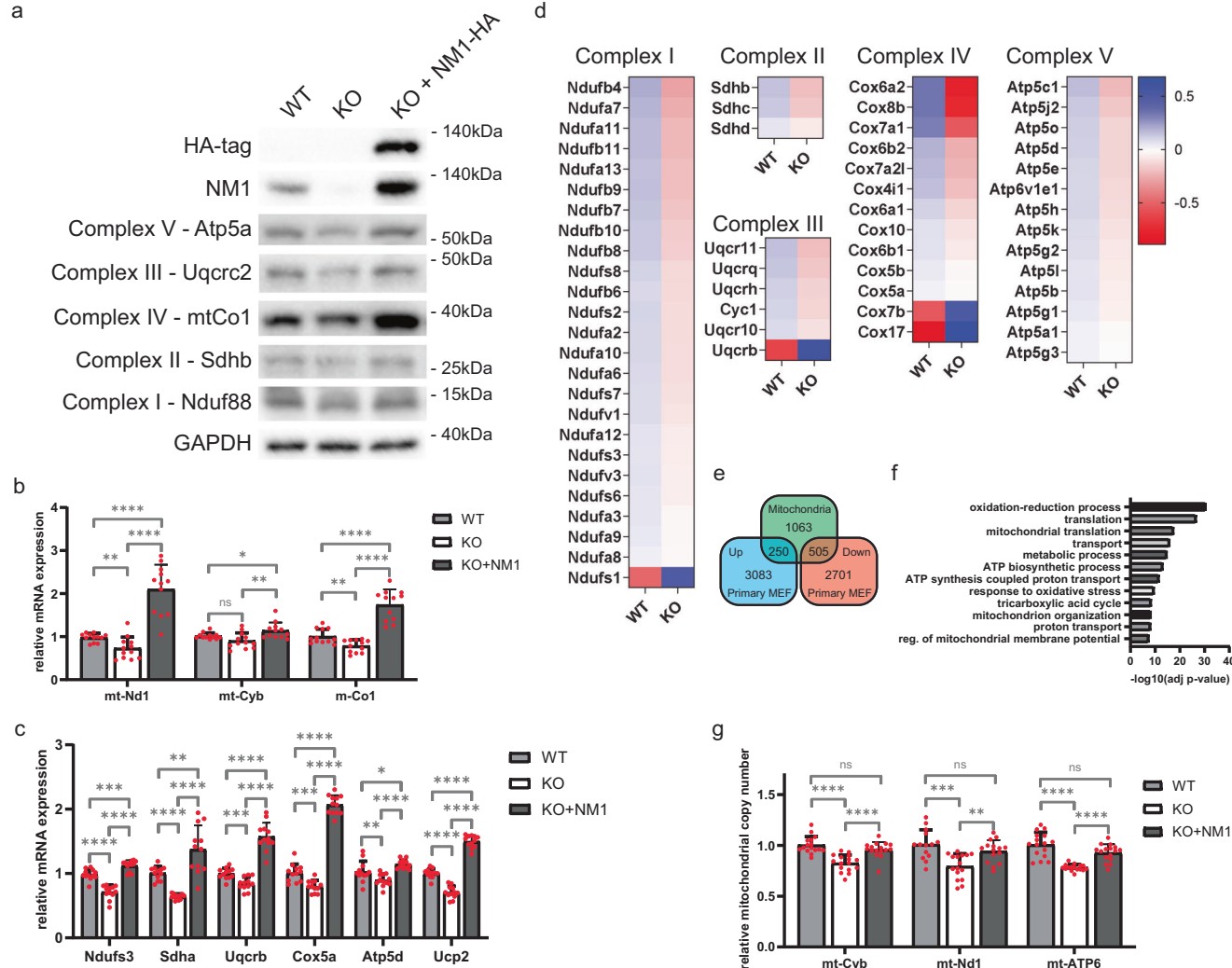

**Fig. 1 | NM1 deletion suppresses the expression of OXPHOS genes and leads to dysregulation of mitochondrial gene expression. a** Western blots from NM1 WT, KO, and KO cells with reintroduced NM1 (KO + NM1-HA) stained with antibodies against proteins in the OXPHOS electron transfer chain, NM1, HA-tag, and control GAPDH. Each western blot was repeated independently three times with similar results. **b** RT-qPCR analysis of mitochondria-encoded genes in WT, KO, and KO + NM1 MEFs. Expression levels are relative to Nono mRNA. An unpaired $t$-test was used for the statistical analysis. $n = 12$, *$p < 0.05$, **$p < 0.01$, ****$p < 0.0001$, ns (not significant). Bars represent mean with error bars representing SD. **c** RT-qPCR analysis of nuclear-encoded mitochondrial OXPHOS genes in WT, KO, and KO + NM1 MEFs. Gene expression levels are relative to Nono mRNA levels. An unpaired $t$-test was used for the statistical analysis. $n = 12$, *$p < 0.05$, **$p < 0.01$, ***$p < 0.001$, ****$p < 0.0001$. Bars represent mean with error bars representing SD. **d** Heatmaps of differentially expressed nuclear OXPHOS genes in primary mouse embryonic fibroblasts isolated from NM1 WT and KO embryos. Genes in the heatmaps are

organized according to their role in the electron transport chain and on their expression profile in the KO cells in ascending order. Plotted values represent the mean of log2-normalized counts for each gene for 3 WT and 3 KO samples. **e** Venn diagram shows the intersection between genes associated with the GO term "Mitochondrion," and all up- and down-regulated differentially expressed genes in primary WT and NM1 KO MEFs. **f** Differentially expressed genes associated with the GO term "Mitochondrion" were subjected to GO analysis, the top 12 enriched biological processes are shown in descending order. **g** qPCR analysis of mitochondrial DNA copy number as determined by the expression of mitochondrial genes mtCyb, mtNd1, and mtATP6 normalized to reads from the nuclear-encoded Terf gene in NM1 WT, KO, and KO + NM1 MEFs. Each dot represents a single measurement. Bars represent mean with SD. An unpaired $t$-test was used for the statistical analysis. $n = 14$, **$p < 0.01$, ***$p < 0.001$, ****$p < 0.0001$, ns (not significant). Source data are provided as a Source Data file.

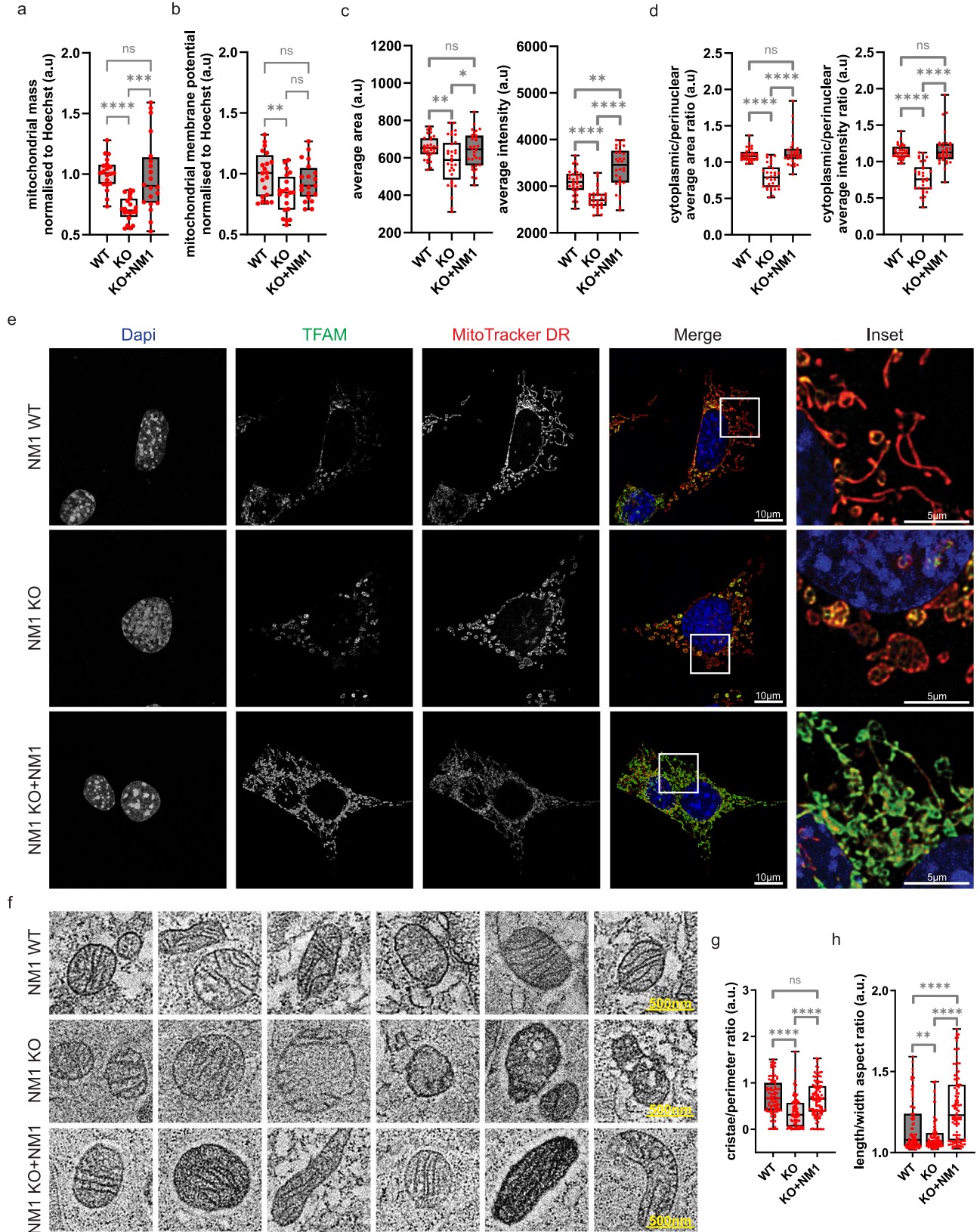

the nuclear region was used for quantification of mitochondrial mass. Results from this analysis are compatible with the above observations and show decreased average intensity and area of Mitotracker DR staining in NM1 KO cells which is fully restored upon NM1 reintroduction in the KO background (Fig. 2c). The mask distinguishing between cytoplasmic and perinuclear regions was next used to define

mitochondrial localization within the cell. Similarly, we found that mitochondria in KO cells are predominantly localized in the perinuclear region while WT and KO + NM1 cells display mitochondria distributed across the entire cytoplasm (Fig. 2d).

To see if decreased mitochondrial mass and membrane potential in NM1 KO MEFs are accompanied by changes in mitochondrial

**Fig. 2 | NM1 depletion leads to reduced mitochondrial mass, perinuclear localization, and altered mitochondrial morphology and structure.**
**a** Spectrophotometric analysis of mitochondrial mass in NM1 WT, KO, and KO + NM1 MEFs. The graph represents the MitoTracker DR fluorescence signal normalized to Hoechst staining in each condition. Each box plot represents the mean value (center line) and first and third-quartile values (box limits). Error bars represent minimum and maximum values. The visualized data represent the compilation of three independent experiments with ≥6 separate measurements for each condition in each experiment. An unpaired $t$-test was used for the statistical analysis. $n = 20$. ***$p < 0.001$, ****$p < 0.0001$, ns (not significant). **b** Spectrophotometric analysis of mitochondrial membrane potential in NM1 WT, KO, and KO + NM1 cells. The graph represents the MitoTracker Orange fluorescence signal normalized to Hoechst staining in each condition. Each box plot represents the mean value (center line) and first and third-quartile values (box limits). Error bars represent minimum and maximum values. The visualized data represent the compilation of three independent experiments with ≥6 separate measurements for each condition in each experiment. An unpaired $t$-test was used for the statistical analysis. $n = 20$. **$p < 0.01$, ns (not significant). **c** High content phenotypic profiling of MitoTracker Deep Red (DR) mitochondrial staining in NM1 WT, KO, and KO + NM1 MEFs. The average mitochondrial area and MitoTracker DR staining intensity per cell are plotted. Each box plot represents the mean value (center line) and first and third-quartile values (box limits). Error bars represent minimum and maximum values. Each dot represents a mean value for one measurement. For each measurement, at least 200 cells have been quantified. An unpaired $t$-test was used for the statistical

analysis. $n = 32$. *$p < 0.05$, **$p < 0.01$, ****$p < 0.0001$, ns (not significant). **d** High content phenotypic profiling of MitoTracker DR mitochondrial staining in NM1 WT, KO, and KO + NM1 MEFs with masks used for differentiating and quantification of perinuclear and cytoplasmic mitochondrial staining. The ratio between cytoplasmic and perinuclear mitochondrial area and intensity of MitoTracker DR stained mitochondria are plotted. Each box plot represents the mean value (center line) and first and third-quartile values (box limits). Error bars represent minimum and maximum values. Each dot represents a mean value for one measurement. For each measurement, at least 200 cells have been quantified. An unpaired t-test was used for the statistical analysis. $n = 32$. ****$p < 0.0001$, ns (not significant). **e** Representative confocal microscopy images of MitoTracker DR and TFAM stained mitochondria in WT, KO, and KO + NM1 MEFs. **f** Representative electron microscopy images of mitochondria in WT, KO, and KO + NM1 MEFs. **g** Quantification of mitochondrial cristae length to mitochondrial perimeter ratio in each condition. Each box plot represents the mean value and first and third quartile values Each box plot represents the mean value (center line) and first and third quartile values (box limits). Error bars represent minimum and maximum values. Each dot represents a single mitochondria measurement. An unpaired $t$-test was used for the statistical analysis. $n = 100$. ****$p < 0.0001$, ns (not significant). **h** Quantification of mitochondrial circularity is defined as length-to-width ratio. Each box plot represents the mean value (center line) and first and third-quartile values (box limits). Error bars represent minimum and maximum values. Each dot represents a single mitochondria measurement. An unpaired $t$-test was used for the statistical analysis. $n = 100$. **$p < 0.01$, ****$p < 0.0001$. Source data are provided as a Source Data file.

morphology, we first visualized WT, KO, and KO + NM1 MEFs by confocal microscopy (Fig. 2e and Supplementary fig. 1). Mitochondria were stained with MitoTracker DR and an antibody against mitochondrial transcription factor A (TFAM), which is the major regulator of mitochondrial copy number and mitochondrial gene expression[29,30]. We found that while WT cells exhibited a network of elongated and circular mitochondria decorated with TFAM staining, typical for differentiated cells[31], NM1 KO cells displayed predominantly circular mitochondria with limited mitochondrial network formation and uneven distribution of TFAM, typical for undifferentiated stem cells, and cancer cells[32,33]. Interestingly, the reintroduction of NM1 in the KO background (KO + NM1) led to increased TFAM levels and the formation of an intricate mitochondrial network (Fig. 2e). To study potential NM1-dependent changes in mitochondrial structure in more detail, we applied transmission electron microscopy (TEM) on WT, KO, and KO + NM1 cells stained with osmium as a contrasting agent to label cellular lipids and mitochondria (Fig. 2f). NM1 KO mitochondria show underdeveloped or missing cristae, more dispersed osmium staining, and abnormalities in mitochondrial shape and inner mitochondrial structure in comparison to the WT. We found that reintroducing NM1 restored cristae formation in these cells as revealed by measuring the ratio between cristae length and mitochondrial perimeter (Fig. 2g) although we could still observe examples of mitochondria with abnormal inner structure similar to KO cells. Measurement of mitochondrial length-to-width ratio shows higher circularity of mitochondria in NM1 KO cells, and prolongation of mitochondria in KO + NM1 cells in comparison to WT condition (Fig. 2h), supporting previous findings obtained by confocal microscopy.

Taken together these results suggest that mitochondrial morphology, function, and distribution are directly dependent on the level of NM1 protein present in cells.

## NM1 depletion results in dysregulated mitochondrial dynamics
To further explore aberrant mitochondrial distribution, structure, and function, we studied gene expression levels of key proteins involved in mitochondrial dynamics. For this purpose, we performed Rt-qPCR analysis on gene markers for mitochondrial fission (DNM1L and Fis1), mitochondrial fusion (Mfn1 and Opa1), mitochondrial quality control (Pink1 and Snca), and mitophagy (Becn and Sqstm). While mitochondrial fission was not affected by NM1 deletion as revealed by the unaltered levels of DNM1L and Fis1 gene expression, all other steps in

mitochondrial turnover were found to be dysregulated, including mitochondrial fusion (increased Mfn1 levels), quality control (decreased expression of both Pink1 and alpha-Synuclein) and mitophagy (decreased Becn levels). NM1 expression in KO cells (KO + NM1) could not rescue the expression of Mfn1 and Becn proteins, while the expression of Pink1 and Snca are partially rescued (Fig. 3a). In addition, fission proteins Fis and Dnm1L whose levels were not changed appeared to be overexpressed in KO + NM1 cells (Fig. 3a). Taken altogether, these findings suggest that in comparison to Oxphos protein expression, NM1 is not directly involved in mitochondrial biogenesis.

## NM1 KO cells switch their metabolism from oxidative phosphorylation to glycolysis
To test whether NM1 KO cells switch from OXPHOS to glycolysis, we first measured the expression of several glycolytic proteins. While some targets did not show altered expression (Hk2, Aldoa, Pkm), glucose transporter Glut1 (Slc2a1), enolase (Eno3), and lactate dehydrogenase A (Ldha), a critical enzyme in pyruvate to lactate conversion, were found to be upregulated in NM1 KO cells while their expression was restored or even reduced to original levels in KO + NM1 cells (Fig. 3b). We next performed fluorometric-based assays to reveal potential differences in mitochondria-related metabolites. Mitochondrial reactive species such as hydrogen peroxide or superoxide ($H_2O_2$) are the main by-products during a series of electron flow processes through an electron transport chain[34]. When measuring the level of $H_2O_2$ we discovered a significant decrease in NM1 KO cells (Fig. 3c) which correlates with a reduction of OXPHOS seen in the transcriptomic analysis. ADP level was not changed between the conditions (Fig. 3d), while ATP level was significantly upregulated (Fig. 3e), which seems contradictory to less energetically profitable glycolysis[35,36] but can be explained by the concomitant increased glucose consumption in the NM1 KO cells measured by fluorescently-labeled deoxyglucose analog 2-NDBG uptake (Fig. 3f). In the final step of glycolysis, pyruvate can be either metabolized in the TCA cycle in mitochondria or can be fermented to lactate and/or lactic acid in glycolysis. While cellular pyruvate level was found to be constant between WT and KO conditions (Fig. 3g), lactate levels were significantly increased in NM1 KO cells (Fig. 3h). Consistently, measurement of intracellular pH by pHrodo green dye showed higher intracellular acidity in the absence of NM1 (Fig. 3i). Apart from the production of ATP and other metabolites, mitochondria serve as the main regulator of calcium homeostasis in

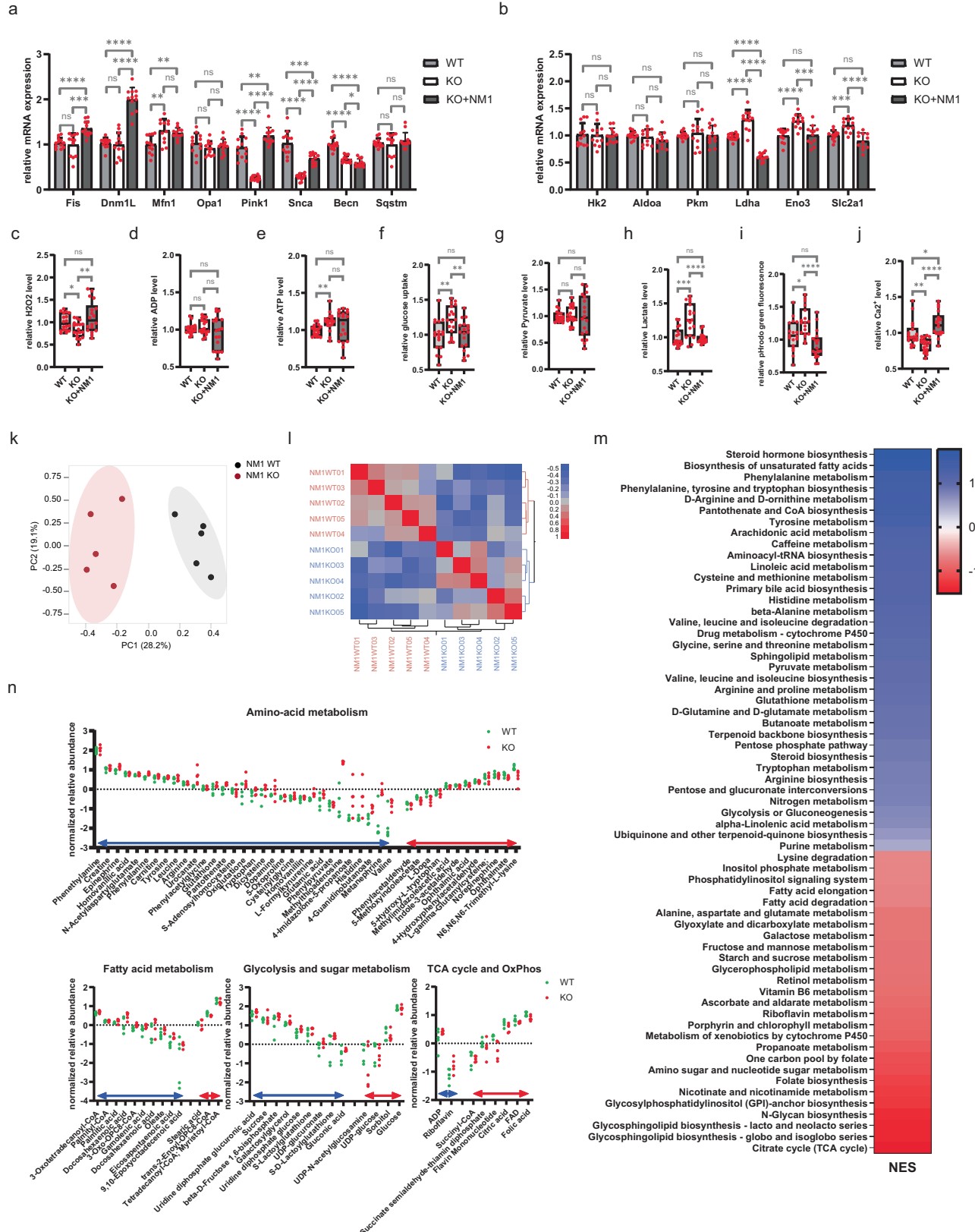

cells through their interactions with other organelles and by their calcium buffering capacity[37]. Therefore, we measured the total amount of intracellular Ca2+ and found calcium levels to be significantly decreased in NM1 KO cells (Fig. 3j). We found that the reintroduction of NM1 in the KO background rescues each phenotype to its original levels but there is higher variability between the samples in comparison to WT condition. These results also suggest that cellular NM1 is likely to require tight regulation for proper mitochondrial function.

Taken together, quantification of mitochondria-related metabolites as well as RTqPCR of glycolytic genes suggest that mitochondrial ATP production and mitochondrial cell signaling are impaired in the

**Fig. 3 | NM1 depletion leads to dysregulated mitochondrial dynamics and a metabolic switch from oxidative phosphorylation to glycolysis. a** RT-qPCR analysis of key genes regulating mitochondrial dynamics in WT, KO, and KO + NM1 cells. The expression of each gene is measured relative to the expression of the Nono gene. Each dot represents a single measurement. Bars represent mean with SD. An unpaired t-test was used for the statistical analysis. $n = 12$, *$p < 0.05$, **$p < 0.01$, ***$p < 0.001$, ****$p < 0.0001$, ns (not significant). **b** RT-qPCR analysis of glycolytic genes in WT, KO, and KO + NM1 cells. The expression of each gene is relative to the expression of the Nono gene. Each dot represents a single measurement. Bars represent mean with SD. An unpaired t-test was used for the statistical analysis. $n = 12$, ***$p < 0.001$, ****$p < 0.0001$. ns (not significant). **c–j** Relative fluorometric quantification of different metabolites in WT, KO, and KO + NM1 cells. Each box plot represents the mean value (center line) and first and third-quartile values (box limits). Error bars represent minimum and maximum values. An

unpaired t-test was used for the statistical analysis. $n = 18$. *$p < 0.05$, **$p < 0.01$, ***$p < 0.001$, ****$p < 0.0001$, ns (not significant). **k** PCA analysis of metabolomic data was performed on the normalized peak dataset from WT and NM1 KO cells. **l** Global metabolomic correlation matrix clustering based on similarity from WT and NM1 KO cells. **m** GSEA analysis of compounds identified using positive ionization organized descending based on their normalized enrichment scores (NES). Blue color represents pathways that are overrepresented and red color pathways that are underrepresented in NM1 KO cells. **n** Graphs represent the normalized relative abundance of each detected compound in the given metabolic pathway. The blue double arrow represents metabolites with relative abundance higher in KO, while the red double arrow represents metabolites with higher abundance in WT. Each dot represents one measurement of a given compound. Source data are provided as a Source Data file.

NM1 KO condition, and upon NM1 deletion cells switch to a glycolytic metabolism even in the presence of oxygen.

## NM1 KO cells exhibit a metabolome profile typical for cancer cells

As the NM1 KO cells seem to move from OXPHOS to aerobic glycolysis[6], we next explored if NM1 depletion leads to metabolic reprogramming. We performed Liquid Chromatography followed by High-Resolution Mass Spectrometry of cellular extracts from NM1 WT and KO cells, and quality control of the peak data resulted in the retention of 7212 from a total of 15,423 compounds for downstream analysis; 4378 compounds were identified using positive ionization and 2839 compounds identified using negative ionization. Principal-component analysis (PCA) of the normalized peak dataset revealed a strong correlation structure in the metabolomic data across both conditions with the first principal component (PC1) capturing the effect of loss of NM1 and showing clear segregation between replicates of the WT and KO. The first two PCs explain 47.3% of the variation in the dataset (Fig. 3k). Clustering of the global metabolomic correlation matrix based on similarity also clearly shows that replicates of each condition cluster together and NM1 deletion leads to distinct metabolomic changes in these cells (Fig. 3l). To identify perturbations in mouse-specific metabolic pathways or metabolite sets in association with the loss of NM1, functional analysis using the Gene Set Enrichment Analysis (GSEA) approach was performed for compounds detected using positive (Fig. 3m) and negative (Supplementary fig. 2) ionization separately and mapped onto *Mus musculus* (mouse) [KEGG]. Among others, the analysis revealed glycolysis, sugar metabolism, amino acid, and fatty acid metabolism pathways to be enriched in the NM1 KO Cells with increased levels of the majority of identified compounds while the TCA cycle and OXPHOS pathways were suppressed in these cells (Fig. 3n). This correlates well with our previous findings and suggests that NM1 KO cells not only switch to aerobic glycolysis but fully reprogram to cancerous metabolism as several recent studies show similar metabolomic profiles in different cancer types[38]. For example, deregulated fatty acid and glucose metabolism were found in lung cancer patients[39]; nucleotide, histidine, and tryptophan metabolism in ovarian cancer[40]; and most prominently, purine metabolism, glycine, serine, arginine and proline metabolism, steroid biosynthesis, sphingolipid metabolism, and bile metabolism deregulated in pancreatic cancer[41]. All aforementioned metabolic pathways were found to be altered in NM1 KO cells as well. A similar analysis was performed where annotated compounds were mapped onto a curated 912 metabolic data sets predicted to change due to dysfunctional enzymes based on human metabolism. The analysis revealed the enrichment of similar pathways as found for the mouse KEGG database (Supplementary data 1).

To test if reintroducing NM1 in the KO background can rescue the metabolomic profile of KO cells, we performed an independent follow-up metabolomic profiling experiment using liquid chromatography

followed by mass spectrometry of cellular extracts prepared from 13 KO, 11 WT, and 15 KO + NM1 biological replicates. Quality control of the peak data resulted in the retention of 891 metabolic features for downstream analysis. Principal-component analysis (PCA) of the normalized peak dataset and hierarchical clustering analysis of all samples showed that WT and KO + NM1 samples have more similar metabolic profiles compared to KO samples (Supplementary fig. 2a, b) and a functional mummichog pathway enrichment analysis highlighted 52 metabolic features with abundance levels that are significantly different in WT and KO + NM1 compared to KO samples (Supplementary fig. 2d, e).

Moreover, we investigated the impact of the NM1 rescue on the abundance of amino acid metabolism and TCA cycle metabolic intermediates. Interestingly, KO + NM1 cells showed a reduction in the levels of metabolites such as betaine, L-isoleucine, and Enol-phenylpyruvate, implicated in amino acid metabolism, to levels that are comparable to that of WT cells relative to KO cells. Of the TCA key intermediates, we highlight succinic acid semialdehyde (SSA) and L-malic acid. KO + NM1 cells showed a reduction in the levels of SSA, which is significantly elevated in the KO cells. Remarkably, SSA is accumulated when the oxidation of succinic semialdehyde to succinic acid is impaired, a key intermediate metabolite of the TCA cycle[42]. KO + NM1 cells also showed rescue of the levels of malic acid, which is a strong indication that the TCA cycle is no longer impaired in the NM1-rescued cells[43] (Supplementary fig. 2e).

In conclusion, global changes in the metabolome of NM1 KO cells associated with increased amino-acid and fatty acid metabolism support the idea of using glycolysis byproducts for rapid biosynthesis of biologically relevant molecules needed for increased proliferation. Herewith associated phenotypes namely decreased production of ROS, imbalance in calcium homeostasis, increased glucose uptake, increased lactate level coupled with increased intracellular acidity, and upregulated expression of glycolytic enzymes, do not only describe characteristics of the glycolytic metabolism but are also the defining hallmarks of cancerous cells and tumors.

## NM1 regulates mitochondria by regulating mitochondrial transcription factors TFAM and PGC1α

To address whether NM1 regulates mitochondrial function via direct interaction with mitochondrial gene promoters, we performed chromatin immunoprecipitation followed by deep sequencing (ChIP-Seq) with antibodies to SNF2H, which binds to NM1 at transcription start sites, and active histone marks H3K9Ac and H3K4me3, which are dependent on PCAF acetylation and Set1b methylation via interaction with NM1. As NM1 is part of the B-WICH remodeling complex, we also examined possible changes in chromatin accessibility by assay for transposase-accessible chromatin with sequencing (ATAC-Seq) in NM1 WT and KO cells. NM1 deletion did not show global changes in SNF2H binding or distribution of active histone marks and did not lead to a change in chromatin accessibility around the transcription start sites

of mitochondrial genes (Supplementary fig. 3a, b). These findings are compatible with the idea that NM1 is not required for SNF2h recruitment but rather for its ATPase activity needed for local nucleosome repositioning[19,44]. They may also reflect the role of NM1 as a general transcription factor rescued by other proteins, as it has been shown for Pol I transcription[26]. Finally, given that NM1 is heavily associated with the non-coding genome, these assays may not be sensitive enough to reflect how NM1 associates with specific transcriptionally active regions. So, we next tested whether changes in gene expression might be a consequence of changes in 3D genome spatial organization. The nuclear genome is hierarchically organized into A and B compartments which respectively correlate with with open and closed chromatin or euchromatin and heterochromatin[45]. The position of the gene in one of these compartments heavily dictates its expression status with genes present in compartment A being transcribed while genes present in compartment B being suppressed. We recently reported that loss of nuclear actin leads to compartment switching, in turn leading to differential expression of genes present within these compartments[46]. As around 80 percent of NM1 DNA binding occurs outside of transcription start sites and gene promoters[20], we next tested whether NM1 could regulate the expression of mitochondrial genes by mediating the 3D spatial organization of the genome rather than via direct regulation of transcription. We, therefore, performed Hi-C chromosome conformation capture analysis combined with deep sequencing (Hi-C-seq) in WT and NM1 KO cells to study compartment changes and their potential effect on mitochondrial gene expression. However, we observed a low level of compartment switching in the NM1 KO condition (Supplementary fig. 3c), and only a very small portion of mitochondrial genes being associated with these regions (Supplementary fig. 3d).

As we were unable to associate NM1 with mitochondrial gene promoters or show the effect of NM1 KO on the 3D reorganization of the genome, we examined whether NM1 could regulate the expression of specific mitochondrial transcription factors, namely Nrf1, Nrf2, PGC1α, Tfb2m, Yy1, Essra and most prominently TFAM[47], as upstream regulators of a broader range of mitochondrial genes. Quantification by RTqPCR confirmed previous results from confocal microscopy showing a reduction of TFAM expression upon NM1 deletion and rescue of TFAM expression upon NM1 reintroduction in the KO background. Similarly, several others were differentially expressed upon NM1 deletion and restored in KO + NM1 cells (Fig. 4a). Based on changes in the expression profiles observed under the different conditions (WT, KO, and KO + NM1), we further focused on two major factors, PGC1α and TFAM. PGC1α is a major mitochondrial co-activator that interacts with the majority of other mitochondrial factors and, as mentioned earlier, TFAM is responsible for transcription of mitochondria-encoded genes and mitochondrial DNA replication[48,49]. We first measured the occupancy of NM1 and histone marks associated with NM1 chromatin-binding – H3K9Ac and H3K4Me3 – at transcription start sites (TSS) of TFAM and PGC1α genes in WT, KO, and KO + NM1 cells by chromatin immunoprecipitation followed by quantitative PCR (ChIPqPCR) (Figs. 4b, c). Results from these experiments show that NM1 binds to TSSs of both genes in WT cells and this correlates with increased H3K9Ac and H3K4Me3 occupancy at TSSs of TFAM and PGC1α. In contrast, in KO cells both histone marks are lost and reintroduction of NM1 in NM1 + KO cells restores both H3K9Ac and H3K4Me3 levels of TFAM and PGC1α TSSs (Fig. 4b, c). These results point towards a direct effect of NM1 in the regulation of active epigenetic marks at the transcription start site of TFAM and PGC1α.

We next hypothesized that if endogenous NM1 is responsible for the expression of OXPHOS genes via regulation of mitochondrial transcription factors, NM1 binding to the transcription start sites of these genes should be reduced once cells switch to glycolytic metabolism. We, therefore, reduced the oxygen level in cells to induce hypoxia-driven glycolysis over OXPHOS, which we have proven by

increased expression of glycolytic genes and decreased expression of OXPHOS genes (Fig. 4d) and performed ChIPqPCR analysis of TSS occupancy of TFAM and PGC1α in WT and KO cells under control (normal oxygen levels) or hypoxic conditions. As expected, NM1 and active histone marks H3K9Ac and H3K4Me3 bind to both TSSs under normoxic conditions, but their binding is heavily decreased upon hypoxia. Similarly, deletion of NM1 leads to a general decrease of histone marks over PGC1α and TFAM TSSs under normal conditions and is even more prominent upon hypoxia (Fig. 4e, f). There is a positive correlation between NM1 binding and the association of active histone marks over TSSs as stronger NM1 binding at the TFAM gene TSS is reflected in the much stronger association of H3K9Ac and H3K4me3 in comparison to PGC1α. Interestingly, even though both histone marks are associated with active transcription, in the case of PGC1α TSS occupancy, NM1 deletion has a much stronger effect on the depletion of H3K9Ac rather than H3K4me3. The difference in the acetylation and methylation patterns between genes could be explained by different PCAF and Set1B associations with assorted transcription factors. For example, PCAF is targeted to DNA not only by NM1 but also by the actin-hnRNPU complex[50] or p53[51] but in both cases, NM1 is part of the complex[23,50]. In contrast, Set1B has been recently shown to be associated with the HIF 1 complex to activate glycolytic genes during hypoxia conditions[52]. This is interesting because at least in colorectal cancer, PGC1a expression is upregulated by hypoxia and leads to increased tumorigenesis[53]. As NM1 deletion leads to the induction of a glycolytic program, it is plausible that loss of H3K4 methylation on PGC1a TSS is partially rescued by glycolytic activation of Set1B. Finally, even though we were unable to find any differences on the global scale by the CHIP-seq and ATAC-seq assays, when we looked at the distribution of ATAC and ChIP-seq signals for SNF2H, H3K9Ac, and H3K4me3 along TFAM and PGC1a we can see a decrease of the majority of marks in NM1 KO cells supporting our ChIPqPCR data (Supplementary fig. 3e, f).

We conclude that NM1 deletion does not affect the expression of mitochondrial genes directly by changing the chromatin landscape or global 3D genome architecture, but rather through direct gene regulation of specific mitochondrial transcription factors – especially TFAM and PGC1α.

## NM1 regulates mitochondrial function via the PI3K/AKT/mTOR pathway and forms a positive feedback loop with mTOR

Extracellular and intracellular nutrient and growth factor sensing and the subsequent metabolic changes that occur are tightly regulated by several mechanisms and pathways. The mTOR kinase as a part of the mTOR complex 1 (mTORC1) controls cellular energetics by regulating transcription and translation of metabolic genes, inducing protein and lipid synthesis upon activation by growth factors through PI3K-AKT signaling[54]. mTORC1 phosphorylates several downstream targets, many of which are mitochondrial transcription factors, which then regulate the expression of mitochondrial genes[55]. Treating cells with specific mTORC1 inhibitor rapamycin leads to reduced expression of PGC1α and other mitochondrial transcription factors and decreased OXPHOS in these cells[56]. As we have seen similar effects on protein expression and OXPHOS in the NM1 KO condition, we next tested whether NM1 could be regulated by the mTORC1 complex. We treated NM1 WT and KO cells with rapamycin and examined NM1 binding to TSSs of mitochondrial transcription factors PGC1α and TFAM by ChIPqPCR. In both cases, rapamycin treatment led to a significant reduction of NM1 occupancy over TSS of both genes followed by reduced distribution of active histone marks (Fig. 4g, h). Interestingly, while in WT cells, rapamycin treatment led to an almost complete drop of active histone marks from TSS, NM1 KO cells seemed to be insensitive to rapamycin treatment, further suggesting the potential importance of an mTOR-NM1 regulatory cascade for cellular metabolism. Since NM1 is directly phosphorylated by GSK3β to stabilize

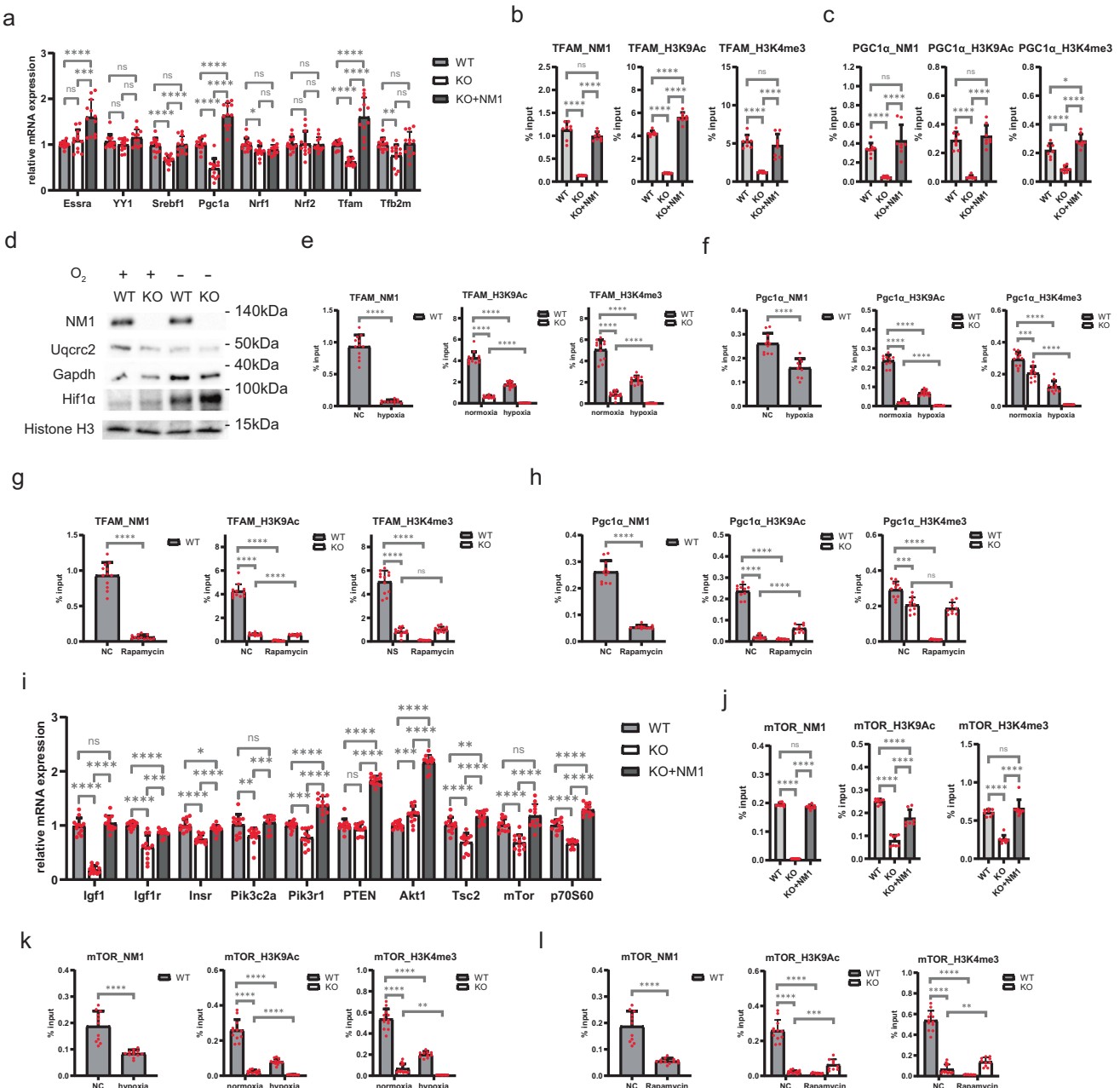

chromatin association and protect it from proteasome degradation[57], we hypothesize that mTOR could be also involved in the regulation of NM1 phosphorylation. Whether the phosphorylation of NM1 by mTORC1 is direct or not needs to be elucidated but we speculate that NM1 could be regulated indirectly via mTOR by GSK3β-dependent phosphorylation and subsequent stabilization of NM1 in the G1 phase of the cell cycle as shown previously[58–60]. We next measured the expression of the main proteins from growth factors to mTOR effector p70S6K and found the whole signaling cascade to be suppressed in the absence of NM1, including mTOR itself, except for Akt1 (Fig. 4i). Based on these observations, NM1 could be a part of a positive feedback loop with mTOR. To test this possibility, we first examined NM1 and active histone mark occupancies at the TSS of the mTOR gene by ChIPqPCR in WT, KO, and KO + NM1 cells. Results from these experiments show that NM1 binds to the mTor TSS and loss of NM1 binding is associated with loss of activating histone marks (Fig. 4j). Similarly, hypoxia (Fig. 4k) and rapamycin treatment (Fig. 4l) led to decreased NM1, H3K9Ac, and H3K4me3 enrichment around the mTOR TSS. This is

further supported by our ATAC and ChIP-seq data showing a decrease of most of the marks at the mTOR TSS in the NM1 KO condition (Supplementary fig. 3g).

Taken together, we suggest a model where NM1 functions as part of a PI3K-AKT-mTOR signaling pathway. Extracellular and intracellular signals activate a cascade of phosphorylation events leading to activation of mTOR which subsequently activates downstream targets and affects NM1 function either directly or indirectly. NM1 then stimulates the expression of mitochondrial transcription factors as well as mTOR itself which leads to multiplication and strengthening of signaling by newly produced mTOR until the extracellular signal persists. This would allow cells to keep relatively small levels of signaling proteins in a resting state and their robust accumulation upon external stimulation.

## NM1 deletion leads to tumorigenesis in mice
We next investigated whether deletion of NM1 is sufficient to induce tumor formation in mice. We injected $3 \times 10^6$ WT, KO, or KO + NM1 cells

**Fig. 4 | NM1 regulates mitochondrial function through the PI3K/AKT/mTOR pathway and forms a positive feedback loop with mTOR. a** RT-qPCR analysis of mitochondrial transcription factors in WT, KO, and KO + NM1 cells. The expression of each gene is measured relatively to the Nono gene expression level. Each dot represents a single measurement. Bars represent mean with SD. An unpaired t-test was used for the statistical analysis. $n = 12$, *$p < 0.05$, **$p < 0.01$, ***$p < 0.001$, ****$p < 0.0001$, ns (not significant). **b** ChIP-qPCR analysis of NM1, H3K9Ac, and H3K4me3 binding to the transcription start site of TFAM normalized to input in NM1 WT, KO, and KO + NM1 cells. Each dot represents a single measurement. Bars represent mean with SD. An unpaired t-test was used for the statistical analysis. $n = 9$, ****$p < 0.0001$, ns (not significant). **c** ChIP-qPCR analysis of NM1, H3K9Ac and H3K4me3 binding to transcription start site of Pgc1α normalized to input in NM1 WT, KO, and KO + NM1 cells. Each dot represents a single measurement. Bars represent mean with SD. An unpaired t-test was used for the statistical analysis. $n = 8$, *$p < 0.05$, ****$p < 0.0001$, ns (not significant). **d** Western blot analysis of cell lysates from WT and NM1 KO cells grown under normoxic conditions ($O_2$ +) or hypoxic conditions ($O_2$ -). Uqcrc2 is part of the OXPHOS pathway, Hif1α and GAPDH are glycolytic genes, and H3 Histone serves as a loading control. Each western blot was repeated independently 3 times with similar results. **e** ChIP-qPCR analysis of NM1, H3K9Ac, and H3K4me3 binding to the transcription start site of TFAM normalized to input under normoxia or hypoxia conditions. Each dot represents a single measurement. Bars represent mean with SD. An unpaired t-test was used for the statistical analysis. $n10$, ****$p < 0.0001$. **f** ChIP-qPCR analysis of NM1, H3K9Ac, and H3K4me3 binding to the transcription start site of Pgc1α normalized to input under normoxia or hypoxia conditions. Each dot represents a single measurement. Bars represent mean with SD. An unpaired t-test was used for the statistical analysis. $n10$, ***$p < 0.001$, ****$p < 0.0001$. **g** ChIP-qPCR analysis of NM1, H3K9Ac, and H3K4me3 binding to the transcription start site of TFAM normalized to input upon Rapamycin treatment. Each dot represents a single measurement. Bars represent mean with SD. An unpaired t-test was used for the statistical analysis. $n = 9$, ****$p < 0.0001$, ns (not significant). **h** ChIP-qPCR analysis of NM1, H3K9Ac, and H3K4me3 binding to the transcription start site of Pgc1α normalized to input upon Rapamycin treatment. Each dot represents a single measurement. Bars represent mean with SD. An unpaired t-test was used for the statistical analysis. $n = 9$, ***$p < 0.001$, ****$p < 0.0001$, ns (not significant). **i** RT-qPCR analysis of PI3K/Akt/mTOR signaling pathway genes in NM1 WT, KO, and KO + NM1 cells. The expression of each gene is relative to Nono gene expression. Each dot represents a single measurement. Bars represent mean with SD. An unpaired t-test was used for the statistical analysis. $n = 12$, *$p < 0.05$, **$p < 0.01$, ***$p < 0.001$, ****$p < 0.0001$. ns (not significant). **j** ChIP-qPCR analysis of NM1, H3K9Ac, and H3K4me3 binding to the transcription start site of mTOR normalized to input in NM1 WT, KO, and KO + NM1 cells. Each dot represents a single measurement. Bars represent mean with SD. An unpaired t-test was used for the statistical analysis. $n = 7$, ****$p < 0.0001$, ns (not significant). **k** ChIP-qPCR analysis of NM1, H3K9Ac, and H3K4me3 binding to TSS of mTOR gene normalized to input in normoxia and hypoxia. Each dot represents a single measurement. Bars represent mean with SD. An unpaired t-test was used for the statistical analysis. $n = 9$, **$p < 0.01$, ****$p < 0.0001$. **l** ChIP-qPCR analysis of NM1, H3K9Ac, and H3K4me3 binding to TSS of mTOR gene normalized to input upon Rapamycin treatment. Each dot represents a single measurement. Bars represent mean with SD. An unpaired t-test was used for the statistical analysis. $n = 9$, **$p < 0.01$, ***$p < 0.001$, ****$p < 0.0001$. Source data are provided as a Source Data file.

in the mammary pads of Balb/c nude mice and monitored tumor formation over 4 weeks. None of the mice with injected NM1 WT cells formed tumors while injection of NM1 KO cells led to rapid tumor growth in all tested animals. Injection of KO + NM1 cells led to an intermediate phenotype when all injected mice formed tumors but its growth and final size were much slower in comparison to tumors formed by KO cells (Fig. 5A–D). Next, we prepared tissue sections from tumors isolated from the mice injected with KO and KO + NM1 cells and tissue sections from the mammary pads of the mice injected with WT cells. Immunohistochemistry analysis with antibodies against cancer markers Bcl-XL, EGFR, and Mct1, showed high positivity in tumor tissues derived from KO cells, unlike the mammary pads of mice injected with NM1 WT cells which don't show any staining and tumors derived from KO + NM1 cells which only show dispersed staining of individual cells (Fig. 5E). Importantly, Mct1 is a monocarboxylate transporter responsible for the shuttling of lactate and pyruvate and serves as a prognostic marker for glycolytic tumors[61]. Overexpression of Mct1 in tumors derived from NM1 KO cells and decrease in its expression in KO + NM1 tumors therefore further supports our previous findings of the metabolic switch from OXPHOS to glycolysis upon NM1 deletion. Hematoxylin and eosin staining of kidney, liver, lungs, and spleen from mice injected with either NM1 WT, KO, or KO + NM1 cells were used for histopathology but the morphology of selected tissues does not show any significant alterations upon injection of each cell type (Fig. 5F). We conclude that NM1 deletion in cells is sufficient for carcinogenesis and formation of solid tumors in the place of injection but at least in this model system, it does not seem to form metastases and secondary tumors in other organs within the duration of the experiment. As WT MEFs did not lead to any tumor growth in Balb/c nude mice, we only used for comparative RNA-seq analysis tumors derived from KO and KO + NM1 cell lines to further elucidate the mechanisms underlying the cellular transformation of NM1 KO MEFs to tumor tissue. Gene expression comparison of the tumors revealed 4714 significantly upregulated and 5581 significantly downregulated genes in tumors derived from KO cells in comparison to tumors derived from KO + NM1 cells. We performed a gene ontology analysis of these genes to reveal the most affected biological processes, cellular components, and pathways between two tumors. The most upregulated GO terms and pathways in KO tumors are associated with cell cycle and cell division, DNA damage signaling and repair, and mRNA processing and transport which correlate with previously published RNA seq data from NM1 KO primary mouse embryonic fibroblasts[23] (Fig. 6a, f, g). In contrast, the most suppressed GO terms and pathways are associated with mitochondrion, aerobic respiration, and oxidative phosphorylation, supporting our previous results (Fig. 6b, h). As tumors derived from KO + NM1 cells show the opposite expression pattern in these pathways we can conclude that NM1 overexpression in tumors can at least partially rescue the main phenotypes observed upon NM1 deletion.

Finally, we studied whether NM1 could play a similar role in tumorigenesis in the context of human cancers. As NM1 is an alternatively spliced variant of *Myo1C* gene, there is no isoform-specific data present in databases, so we first looked at the mutagenesis rate (Fig. 7a) and deregulation of gene expression (Fig. 7b) of *Myo1C* compared to *p53*, and *mTOR* in human cancer samples collected in the COSMIC Catalogue of Somatic Mutations in Cancer (version COSMIC v96)[62]. While the *p53* gene shows a very high mutagenic rate in the majority of cancers, *Myo1C* and *mTOR* genes show relatively low levels of mutagenesis in different cancer tissues (Fig. 7a). However, gene expression analysis shows that Myo1C is predominantly downregulated in several types of cancer, with ovarian cancer being the most prevalent (50% of all cancer cases have reduced expression of Myo1C), followed by the large intestine, kidney, lung, urinary tract, and breast cancers. p53 protein as a tumor suppressor, shows a similar pattern to Myo1C with reduced expression in the majority of cancers, while mTOR as an oncogene is predominantly overexpressed in most of the cancer tissues (Fig. 7b). To find out if mutations in the *Myo1C* gene in human cancers are associated with pathways known to be affected in NM1 KO mouse embryonic fibroblasts, we next performed single sample gene set enrichment analysis followed by linear regression analysis and found that proliferation, apoptosis, and most importantly glycolysis pathways are upregulated in cancers with *Myo1C* mutations while mTOR pathway is suppressed in these cancers (Fig. 7c, d).

We conclude that in the mouse model system, NM1 serves a role as a tumor suppressor and its deletion leads to changes in cell metabolism that are directly connected to tumorigenesis, which could have similar effects in human tumors.

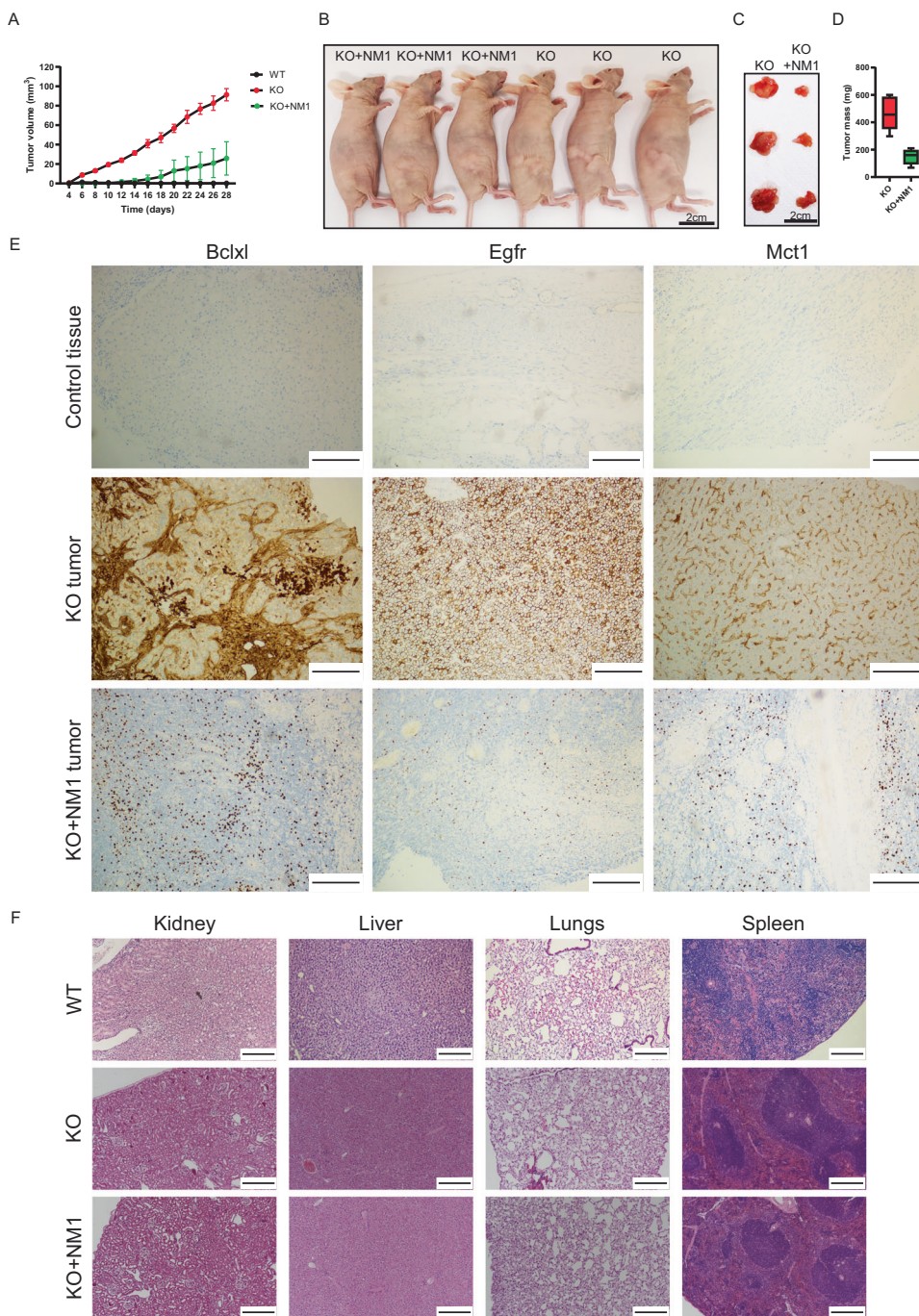

**Fig. 5 | NM1 is a potential tumor suppressor. A** The tumor growth rate over 28 days, in Balb/c nude mice injected with either WT, KO, or KO + NM1 cells. Bars represent minimal and maximal values. $n = 5$. **B** Representative pictures of Balb/c nude mice with developed tumors 28 days after injection of KO or KO + NM1 cells in mammary pad tissue. **C** Representative tumor tissues isolated from Balb/c nude mice 28 days after injection with KO and KO + NM1 cells. **D** Mass measurements of tumors isolated from KO and KO + NM1 cell-injected Balb/c nude mice. Each box plot represents the mean value (center line) and first and third-quartile values (box limits). Error bars represent minimum and maximum values. $n = 5$. **E** Immunohistochemistry staining with antibodies against cancer markers Bclxl, Egfr, and Mct1 of control tissue isolated from the place of injection of NM1 WT cells and tumor tissue isolated from mice injected with either NM1 KO cells or KO cells with reintroduced NM1 (KO + NM1). Each antibody staining was repeated independently 3 times with similar results. Scale bar = 250 μm. **F** Staining of sections of vital organs from mice injected with NM1 WT, KO, or KO + NM1 cells. Each antibody staining was repeated independently three times with similar results. Scale bar = 250 μm. Source data are provided as a Source Data file.

## Discussion

In the present study, we report that nuclear myosin 1 directly regulates the expression of specific mitochondrial transcription factors and forms a regulatory feedback loop with upstream signaling protein mTOR. Cells lacking NM1 show suppressed PI3K/AKT/mTOR signaling pathways and reduced expression of mitochondrial transcription factors. This leads to mitochondrial phenotypic changes associated with a metabolic switch from OXPHOS to aerobic glycolysis which may serve as a potential underlying mechanism for solid tumor formation.

As the translation of new proteins is an energetically heavy process, cells tend to keep signaling molecules to a minimum during starvation. In response to external stimuli, phosphorylation cascades

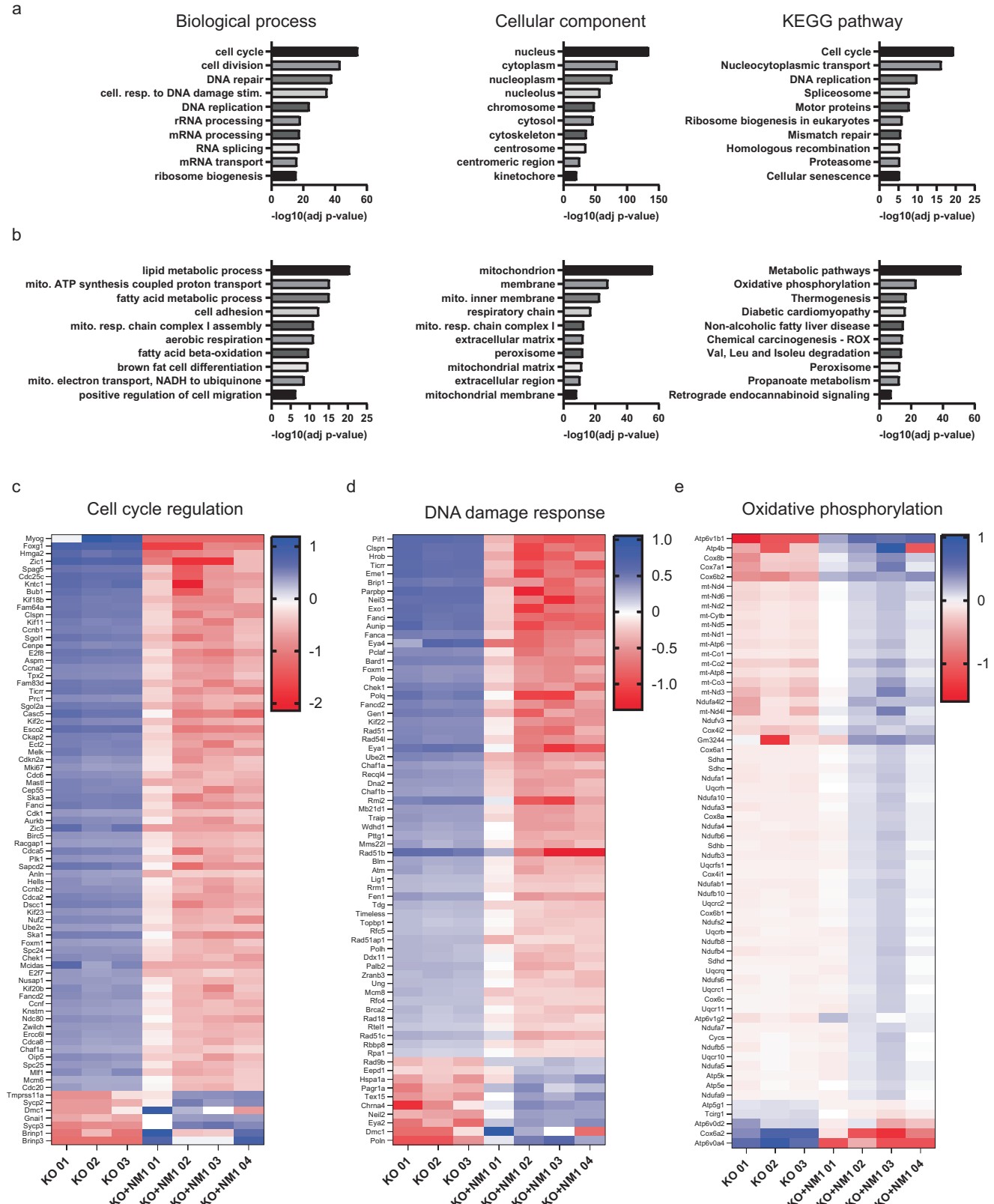

are used to amplify the signal leading to a fast and adequate response. As the signal persists over time, cascades get saturated and cells cannot be further stimulated. Therefore, every pathway has several positive and or negative feedback loops which allow for fast adjustment of a given pathway to cellular needs. We provided evidence that NM1 is a key element involved in such a positive feedback loop with mTOR and

it is, therefore, important for proper cell signaling in cells. In response to stimuli, the PI3K/AKT/mTOR pathway is activated leading to the binding of NM1 to promoters of mitochondrial transcription factors and also mTOR itself and their subsequent robust expression This leads to increased mitochondrial biogenesis and the accumulation of mTOR which can activate more NM1, potentially until a stimulus

**Fig. 6 | Transcriptomic profiling of tumor tissues derived from KO and KO + NM1 MEFs shows transcriptional rescue upon NM1 overexpression. a** Gene ontology analysis of genes that are found to be upregulated in tumors derived from KO cells in comparison to tumors derived from KO + NM1 cells. The top 10 enriched biological process, cellular component, and KEGG pathway terms are shown in descending order based on their significance. **b** Gene ontology analysis of genes that are found to be downregulated in tumors derived from KO cells in comparison to tumors derived from KO + NM1 cells. The top 10 enriched biological process, cellular component, and KEGG pathway terms are shown in descending order based on their significance. **c** Heatmap of most differentially expressed genes (log2FC ≥ 4 for upregulated genes and log2FC ≤ -4 for downregulated genes) between tumors associated with cell cycle and cell division. Heatmap represents

the log2-normalized counts for each tumor sample derived either from KO or KO + NM1 cells organized in descending order based on log2 fold change value. **d** Heatmap of the most differentially expressed genes (log2FC ≥ 2 for upregulated genes and log2FC ≤ -2 for downregulated genes) between tumors associated with DNA damage response and repair. Heatmap represents the log2-normalized counts for each tumor sample derived either from KO or KO + NM1 cells organized in descending order based on log2 fold change value. **e** Heatmap of differentially expressed genes (log2FC ≥ 1 for upregulated genes and log2FC ≤ -1 for down-regulated genes) between tumors associated with Oxidative phosphorylation. Heatmap represents the log2-normalized counts for each tumor sample derived either from KO or KO + NM1 cells organized in descending order based on log2 fold change value. Source data are provided as a Source Data file.

---

persists. After the initial signal is lost, the phosphorylation cascade comes to a halt, NM1 does not activate mTOR expression any further and the mTOR protein levels return to basal levels (Fig. 8). Interestingly, except for Akt kinase, the whole PI3K/Akt/mTOR signaling cascade is suppressed in NM1 KO cells, while previous studies suggested that upon inhibition of mTOR, PI3K, and AKT pathways are activated via insulin-like growth factor receptor alternative pathways[63,64]. This partially correlates with the observed increase in Akt kinase expression in NM1-depleted cells, however, insulin-like growth factor/receptor, as well as members of the PI3K pathway, are suppressed suggesting some other mechanism is responsible for Akt kinase activation and suppression of PI3K pathway. An explanation could be that NM1, similarly to mTOR, transcriptionally regulates the expression of these genes. However, there is a possibility that NM1 has a direct effect on these proteins as NM1 is directly associated with the plasma membrane-bound phospholipids and regulates plasma membrane dynamics and organization[26,65,66]. For example, Myo1C has been shown to facilitate exocytosis and delivery of several proteins such as Glut4, Neph1, aquaporin2, or VEGFR2 receptor to the plasma membrane and it is plausible that deletion of NM1 would affect the distribution/expression of plasma membrane proteins such as IGF1R as well[67–70]. Another possibility for NM1 regulation of the PI3K pathway could be via competitive binding to phosphatidylinositol (3,4)-bisphosphate (PIP2). PI3K phosphorylates membrane-bound PIP2 to phosphatidylinositol (3,4,5)-trisphosphate (PIP3) and the balance between PIP2 and PIP3 is critical for cellular homeostasis[71]. Dysregulation of lipid signaling could therefore affect the PI3K levels as well. Both NM1 and Myo1C were shown to specifically bind to PIP2 at the plasma membrane and we showed previously that upon deletion of NM1, the amount of myosin molecules bound to PIP2 is reduced by half[66,72]. The abundance of free PIP2 molecules and lack of competition between plasma membrane-bound NM1 and PI3K could lead to an adjustment of PI3K expression for proper signaling and cellular homeostasis.

The signaling network defined by PI3K, AKT, and mTOR proteins controls several essential biological functions such as cellular growth, cell metabolism, and survival, and as a pro-proliferative pathway is often deregulated in cancers. It is therefore interesting that upon deletion of NM1, cells proliferate faster and can form tumors in a nude mouse model even though the PI3K/AKT/mTOR signaling pathway is suppressed. This could be at least partially explained by the previously described role of NM1 in cell cycle regulation[19,23] although the possibility that NM1 affects other signaling pathways cannot be excluded. In favor of this, several studies suggested that even though often upregulated in cancers, targeting the mTOR pathway with its inhibitors brings only poor outcomes due to the promiscuity of signaling cascades and the plethora of possible targets that can be activated/deactivated depending on the microenvironment[73]. Apart from insulin-like growth factor receptor-driven activation of PI3K and Akt mentioned above[63,64], activation of other pathways such as DNA-PK or MAPK/ERK or inactivation of GSK3-dependent proteasomal degradation of oncogenic proteins can promote cancer cell survival even upon mTOR suppression[74–77]. Accordingly, mTOR itself can be regulated via

canonical AKT pathway or by Akt-independent Adenosine Monophosphate-activated Protein Kinase (AMPK) pathway[78], and depending on conditions, mTOR can promote transcriptional programs for both, OXPHOS and glycolysis[79–81].

Another explanation for rapid tumor growth caused by NM1 KO cells can be due to a change in the tumor microenvironment. Several studies have indeed shown that metabolic resetting has an early active role in cellular reprograming into pluripotent or cancer stem cells and only after the initial switch to stemness metabolism can pluripotency transcriptional regulators induce additional factors to achieve stemness[82–88]. Additionally, hypoxia, low nutrient content, and increased acidity due to an abundance of lactic acid not only affect cancer/stem cell metabolism but also influence surrounding cells and tissues which can further promote tumorigenesis[89,90]. Similarly, early epigenetic changes leading to the downregulation of differentiation programs are a prerequisite for a metabolic switch and achievement of stemness[83], and suppressing mTOR in a timely manner by stemness and oncogenic transcription factor Sox2 is needed for stemness acquisition[91]. As we have shown that loss of NM1 suppresses the global level of active histone marks associated with gene promoters and transcription start sites, increases levels of heterochromatin histone marks[23], and its deletion leads to a metabolic switch from OXPHOS to glycolysis, combination of these factors in NM1 KO cells may provide sufficient signal for cell transition to cancer gene programs even though they have suppressed PI3K/Akt/mTOR signaling pathway. We speculate that this can serve as a rescue mechanism for cancer cells during mTOR-targeted cancer therapy as Rapamycin treatment leads to decreased binding of NM1 and active histone marks around transcription start sites of mitochondrial transcription factors similar to NM1 KO cells. The question of whether suppressing OXPHOS by NM1 depletion is sufficient for the metabolic switch to glycolysis by itself, or whether some other pro-glycolytic pathways must be activated remains to be elucidated. However, suppression of p53 protein was shown to induce expression of glucose transporters Glut1 and lactate/pyruvate transporter Mct1 leading to increased glucose metabolism and glycolysis[92,93] and we have shown that NM1 regulates the expression of p21 in cooperation with p53[23]. As we have shown here that NM1 deletion leads to upregulation of Glut1 and Mct1 both in cells and tumors, it is plausible that deletion of NM1 not only directly suppresses OXPHOS via mitochondrial transcription factors but could also promote glycolysis via affecting the expression of some p53 target genes.

Glycolysis serves as a primary source of energy production in pluripotent and cancer stem cells and switching to OXPHOS is one of the defining hallmarks of gradual differentiation. NM1 deletion could help to preserve the stemness of pluripotent stem cells as several studies showed that forced expression of glycolytic enzymes in pluripotent cells could protect them from differentiation[94]. On the other hand, targeted activation of NM1 could push cells toward OXPHOS preventing the onset of cancer development or in the differentiation and maintenance of specific cell types such as neurons which are heavily dependent on OXPHOS. This is especially interesting as we have shown that overexpression of NM1 in KO cells leads not only to a

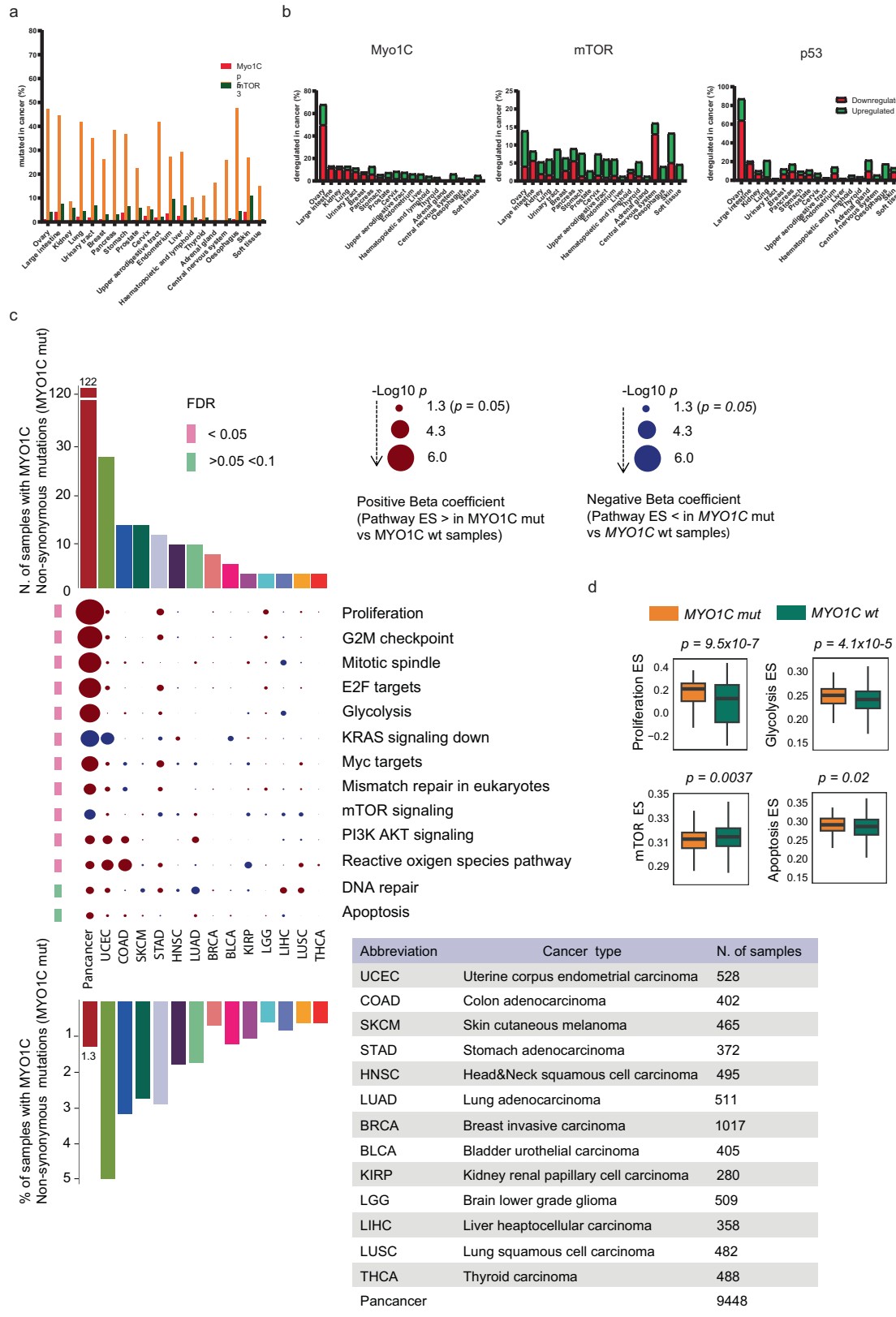

rescue of the metabolic phenotypes but even to their further stimulation. While elucidating how much NM1 and when exactly is enough to cause certain phenotypes remains to be investigated, taken altogether our results suggest a role for NM1 as a tumor suppressor through a mechanism regulating mitochondrial transcription factors and subsequently oxidative phosphorylation.

## Methods

Performed research complies with all relevant ethical regulations and was approved by NYU Institutional Biosafety Committee (project 327 - The role of nuclear myosin 1 in differentiation and disease). All animal experiments were performed after approval by the NYUAD-IACUC (Protocol 21-0005).

**Fig. 7 |** *Myo1C* **mutagenesis-association in human cancers and analysis of dysregulated oncogenic pathways in** *Myo1C***-mutated human cancers. a** The mutagenesis rate of *Myo1C*, *p53*, and *mTOR* in different human tissue cancer samples based on the COSMIC database. **b** The gene expression rate of Myo1C, p53, and mTOR in different human tissue cancer samples based on the COSMIC database. **c** Heatmap displaying the associations between *Myo1C* non-synonymous somatic mutations and the enrichment score (ES) of oncogenic pathways across TCGA cancers, combined (Pan-cancer) and stratified per cancer type (Per-cancer). The p values are derived from linear regression models. Pathways with a significant association in the Pan-cancer analysis (FDR < 0.1) are displayed. Beta coefficients above 0 (red) indicate a positive association between the pathway ES and the presence of *Myo1C* non-synonymous mutations; Beta coefficients below 0 (blue)

represent a negative association. The size of the radius represents the negative Log10 p value. The number and proportion of samples harboring *Myo1C* non-synonymous mutations are represented by Pan-cancer and Per-cancer. Tumor types having less than three samples harboring *Myo1C* non-synonymous mutations are omitted in the Per-cancer analysis but retained in the Pan-cancer analysis. **d** Box plots showing the enrichment scores of four representative oncogenic pathways in Myo1C mutated and non-mutated groups in pan-cancer samples. Centerline, box limits, and whiskers represent the median, interquartile range, and 1.5x interquartile range. The p-values are derived from linear regression analyses. An unpaired t-test was used for the statistical analysis. $n = 122$ for Myo1C mutated samples, $n = 9326$ for Myo1C non-mutated samples. Source data are provided as a Source Data file.

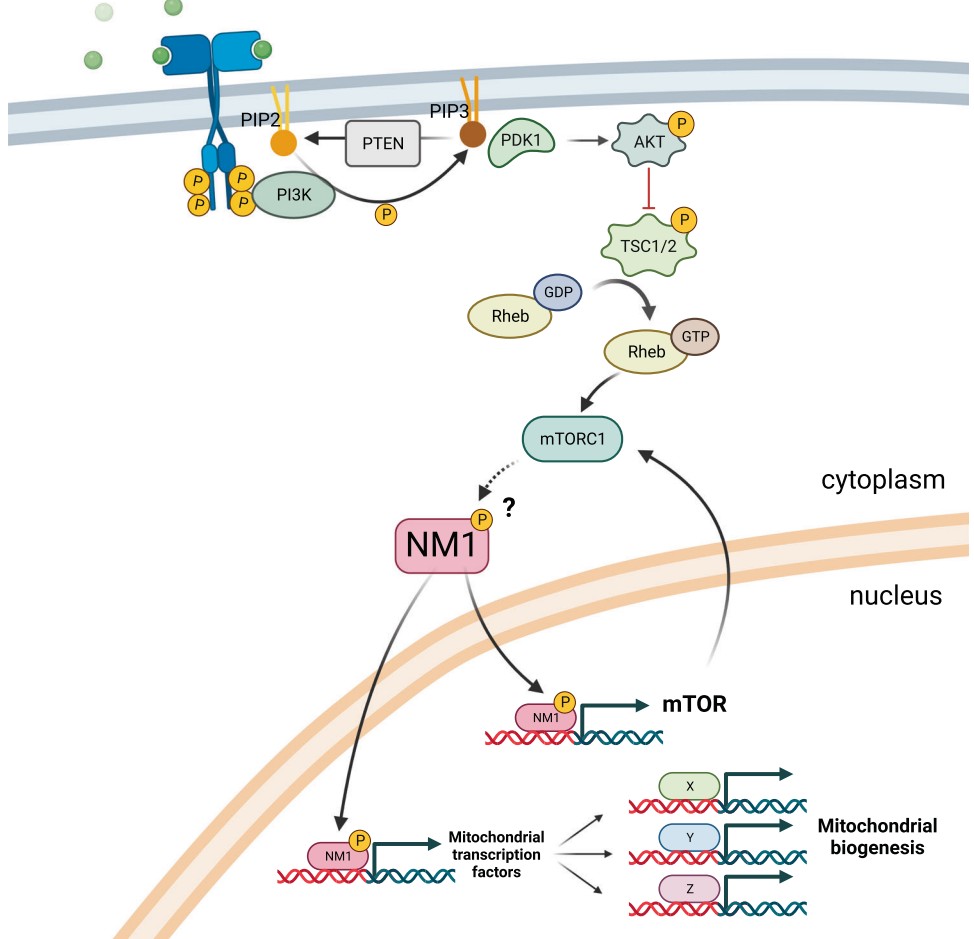

**Fig. 8 | Proposed mechanism of NM1 action in PI3K/Akt/mTOR signaling.** We propose that NM1 has a critical function as a regulator of cellular metabolism and it is part of the PI3K/Akt/mTOR pathway. Upon external stimulus, PI3K/Akt/mTOR cascade is activated leading to phosphorylation of several downstream targets activating NM1 which in a positive feedback loop transcriptionally regulates mTOR and the expression of mitochondrial transcription factors TFAM and PGC1α responsible for mitochondrial biogenesis. Created with BioRender.com.

## Cell culture, reagents, and antibodies

Nuclear Myosin 1 Knock-Out (NM1 KO) cell lines were derived from wild-type mouse embryonic fibroblasts (MEFs) (ATCC® CRL-2752) (NM1 WT) using the CRISPR/Cas9 system[23]. For the NM1 rescue experiment, we transduced NM1 KO MEFs by lentiviral vector carrying coding sequence for NM1 fused with HA and V5 tag (VectorBuilder). The cells expressing NM1-HA-V5 were selected by Neomycin and single clones were tested by Western blot by specific anti-HA antibodies. Cells were grown in a DMEM medium containing 10% fetal bovine serum, 100 U/ml penicillin, and 100 mg/ml streptomycin (Millipore-Sigma) in a humidified incubator with 5% $CO_2$ at 37 °C. Hypoxia

experiments were performed in a nitrogen-pressured incubator to keep stable 5% $CO_2$ and 1% $O_2$ conditions for 48 h before subsequent experiments. For drug treatments, cells were incubated with 100 nM Rapamycin in a normal full DMEM medium for 2 h to specifically inhibit the mTorc1 complex.

Antibodies against GAPDH (ab8245), The Total OXPHOS Rodent WB Antibody Cocktail kit (ab110413), H3K9Ac (ab4441), H3K4me3 (ab8580), HA-tag (ab9110), Snf2h (ab3749), Alexa Fluor 555 Goat Anti-Rabbit (ab150078), Horseradish peroxidase (HRP)-fused Goat Anti-Rabbit (ab6721) and Rabbit Anti-Mouse (ab6728) were purchased from Abcam. Antibodies against TFAM (ABE483) and Mct1

(AB3538P) were purchased from Merc Millipore. Antibody against EGFR (2–18C9) was obtained from Agilent. The anti-Hif1a (D1S7W) antibody was obtained from Cell signaling. The anti-beta actin (A5316) antibody was obtained from Sigma. The anti-NM1 antibody was described and characterized earlier[19,20,57,95]. Where applicable, all antibodies were used according to manufacturers' protocols. Primary antibodies were diluted 1:200 and secondary antibodies 1:400 in 1x PBST buffer with 3% bovine serum albumin for immuno-fluorescent experiments, and 1:1000 and 1:2000 respectively in 1xTBST buffer for western blot analysis. MitoTracker™ Deep Red (M22426) and MitoTracker™ Orange (M7510) were purchased from ThermoFisher Scientific. Hoechst 43222 (H1399) and ProLong Gold Antifade Mountant with 4',6- diamidino-2-phenylindole (DAPI; P36931) were purchased from Invitrogen.

## High-content phenotypic profiling

384-well clear-bottom assay plates (Corning) were used to culture cells at a density of 500–1000 cells per well for the high-content phenotypic profiling. Cells were incubated with 200 nM MitoTracker™ Deep Red and Hoechst (4 µM) for 30 min, washed twice with 1xPBS, fixed in 4% formaldehyde for 10 min and stored in 1xPBS buffer. The plate was analyzed via the Cellomics ArrayScan XTI High-Content Screening platform (ThermoFisher Scientific), and image analysis was performed using the Compartment Analysis BioApplication software (Thermo Fisher Scientific). Primary objects (Circ) were defined using the Hoechst stained nuclei. Secondary masks were used to define mito-chondria as "spots" with a positive fluorescent signal. Based on the analysis, the average fluorescence intensity or area of these spots was measured to quantify mitochondrial mass and distribution within the cell. For each experiment, 24 wells per condition, each containing at least 250 cells were used and obtained data were plotted as a mean value for each well.

## Western blotting

RIPA buffer (50 mM Tris-HCl pH 7.5, 150 mM NaCl, 1 mM EDTA, 1 % NP−40, 0.5% sodium deoxycholate, and 0.1 % SDS) containing 1x cOm-plete protease inhibitor cocktail (Roche) was used to collect total cellular lysates from MEFs. Pierce BCA protein assay kit (Thermo-Fisher Scientific) was used to measure protein concentration in all samples. 20 µg of total protein per sample together with 1x Laemmli buffer was loaded to a 10% SDS-polyacrylamide gel electrophoresis (PAGE) gel and separated under reducing conditions. The separated proteins were transferred to a polyvinylidene difluoride membrane, after which the membrane was blocked with 3% milk in 1x TBST buffer (20 mM Tris pH 7.5, 150 mM NaCl, 0.1% Tween 20) for 1 hr. Immunoblotting with primary antibodies was performed overnight at 4 °C in a rotator, followed by 3 washes with 1x TBST, subsequent incubation with HRP-fused secondary antibodies for 4 h at 4 °C in a rotator, and 3 washes with 1xTBST. Protein bands were developed with ECL Western Blot Substrate (BioRad) and western blots were imaged by a ChemiDoc MP Imaging system (BioRad). The quantifi-cation of protein bands was performed by ImageJ software by com-paring the signal from each antibody to the GAPDH signal which served as a loading control. Four blots were used for the quantifi-cation of each sample.

## Quantitative RT-PCR

RNazol (Sigma) was used to extract total RNA from NM1 WT and KO MEFs according to the manufacturer's protocol. The extract was cleaned from residual gDNA with a Turbo DNA-free kit (Ambion). 1ug of RNA per sample was reverse transcribed using RevertAID First Strand cDNA Synthesis Kit (ThermoFisher Scientific) and diluted in water. The resulting cDNA was used as a template for quantitative PCR by Maxima SYBR Green qPCR Mix (ThermoFisher Scientific) with relevant primers for selected genes (Supplementary data 2). Three-

step cycling protocol was used to amplify the signal in the StepOnePlus Real-Time Thermal Cycler (Thermo Fisher Scientific). The expression data for each sample was normalized to Non-POU Domain Containing Octamer Binding (Nono) protein expression[96]. qPCR analysis for each gene was performed in triplicates and final results were compiled from at least 3 independent experiments.

## Chromatin immunoprecipitation and qPCR

Initial chromatin immunoprecipitation (ChIP) analysis was per-formed on WT, KO, and KO + NM1 cells grown under normal condi-tions. Subsequent experiments were performed on WT, and KO cells specifically under normoxia or hypoxia conditions, or treated with mTorc1-inhibitor Rapamycin. Approximately 5 million cells per ChIP were fixed with 1% formaldehyde, followed by 10 min of incubation with 0.125 M glycine to stop the reaction. Cells were lysed with the lysis buffer (10 mM Tris pH 8.0, 10 mM NaCl, 0.2% NP-40) supple-mented with protease inhibitors, and nuclei were recovered and resuspended in nuclei lysis buffer (50 mM Tris pH 8.1, 10 mM EDTA pH 8.0, 1% SDS, ddH2O) with protease inhibitors. Chromatin was sonicated by QSONICA sonicator using 4 rounds of DNA shearing on ice with a probe submerged into the bottom of the sample (70% Amplitude, 1 sec on/1 sec off for 5 min). The final DNA fragment size was checked by DNA electrophoresis to be 200–500 bp. Sheared chromatin was divided equally for immunoprecipitation with anti-bodies fused to Dynabeads (Thermo Fisher Scientific) in IP Dilution buffer (20 mM Tris pH 8.1, 2 mM EDTA pH 8.0, 150 mM NaCl, 1% Triton X-100, 0.01% SDS, ddH2O). 10 µg anti-NM1, 2 µg anti-H3K9Ac, and 2 µg of anti-H3K4me3 antibodies were used in each ChIP condi-tion. 10% of sheared chromatin served as input control. Samples were incubated overnight, rotating at 4 °C. The IP samples were washed with IP Wash buffer 1 (20 mM Tris pH 8.1, 2 mM EDTA pH 8.0, 50 mM NaCl, 1% Triton X-100, 0.1% SDS, ddH2O) and IP Wash buffer 2 (20 mM Tris pH 8.1, 1 mM EDTA pH 8.0, 0.25 M LiCl, 1% NP-40, 1% sodium deoxycholate monohydrate, ddH2O). Reverse crosslinking and elution of DNA were performed by using the elution buffer (100 mM NaHCO3, 1% SDS, ddH2O). The samples were incubated with 5 M NaCl and 10 mg/ml of RNase A at 65 °C for 1 h. Proteinase K (20 mg/ml) was then added and incubated at 55 °C for 2 h. The samples were then placed on a magnet and the immunoprecipitated DNA samples in the supernatant along with the input samples were purified using ChIP Purification Kit (Zymo Research), according to the manufacturer's protocol. They were diluted in 12 µl of elution buffer and the concentration was measured by Qubit (Thermo Fisher Scientific).

qPCR analysis was performed on a diluted immunoprecipitated sample or input control in each reaction mixed with Maxima SYBR Green/Rox qPCR Master mix (Thermo Fisher Scientific) and appro-priate set of primers (Supplementary Data 2), followed by a three-step cycling protocol in the StepOnePlus Real-Time Thermal Cycler (Thermo Fisher Scientific). Each graph represents combined data from 3 biological replicates, and each performed at least in 2 tech-nical replicates, and each sample was normalized to the adjusted input.

## Mitochondrial mass and mitochondrial membrane potential measurement

To analyze differences in mitochondrial mass and membrane potential between NM1 WT and KO cells were seeded in 96 well plates (20,000 cells/well) and incubated with Hoechst (4 µM) (Ex/Em 361/497) and 200 nM MitoTracker™ Deep Red (Ex/Em 644/665) or 100 nM Mito-Tracker™ Orange (Ex/Em 554/576) respectively, for 30 min. Cells were washed with 1xPBS twice, and the total fluorescence signal for each dye was measured with Synergy H1 Hybrid microplate Reader (BioTek). The signal obtained from the Mitotracker dyes was normalized to Hoechst staining. The visualized data represent the compilation of 3

independent experiments with 6 wells separately measured for each condition in each experiment.

## Microscopic methods

Cells grown overnight on glass slides were stained with 200 nM MitoTracker™ Deep Red FM for 20 min. Cells were washed twice with 1xPBS, fixed with 4% formaldehyde, permeabilized with 0.5% Triton X-100 in PBS for 15 min, and blocked with 1% BSA for 1 h. This was followed by staining with primary antibodies against TFAM (1:200) overnight. Cells were washed thrice with 1xPBST buffer and stained with Alexa Fluor 555 Goat Anti-Rabbit antibody (1:400) for 4 h. After 3 additional washes with 1xPBST buffer, ProLong Gold anti-fade mounting media with DAPI (Invitrogen) was used to mount the cover-slips to a glass slide. The Leica TCS SP8 STED 3X microscope equipped with HyD SMD2 detector and Leica HCPL APO CS2 63x/1.4 oil objective was used to acquire confocal images, the Software Leica application SuiteX was used to capture and analyze images, and the Huygens Professional software was used for deconvolution. Final data were processed using the Fiji software.

## Transmission electron microscopy sample preparation

NM1 WT, KO, and KO + NM1 cells were washed twice with PBS and trypsinized and 10 million cells for each genotype were collected and spun down at low speed to form a firm pellet. High-pressure freezing was performed using the Leica ICE high-pressure freezer apparatus with a gold-plated specimen carrier (carrier A, 3 mm diameter, 100 μm deep, and carrier B flat side down). Specimen carriers were lightly coated with 0.1% soy lecithin in chloroform to ensure a smooth and easy opening of the carrier without damaging the pellet. Carrier A was filled with well-pelleted cells and carrier B was placed on top with the flat side down before freezing at a programmed pressure of 2100 bars. After freezing, the sample pod was released automatically into a liquid nitrogen bath; the sample carrier was then separated from the specimen pod using precooled fine-tipped tweezers under liquid nitrogen and transferred to Leica freeze-substitution AFS2 set up in a 2 mL solution of cold dry absolute acetone (v/v) containing 1% osmium tetroxide. The AFS unit was slowly warmed from -90 °C to 0 °C (2 °C/h), with the temperature being held at both −90 °C for a period of 15hrs and thereafter at −60 °C and −30 °C for a period of 8hrs each. Samples were cleared of osmium by rinsing with absolute acetone (3 times × 5 mins) and thereafter infiltrated with low-viscosity resin with increasing concentrations of 30% and 66% for 4 h each, and 100% overnight. Individual samples were embedded in 1 mL of 100% low-viscosity resin and polymerized for 30 h at 60 °C. The resin blocks were tapered into a pyramid shape and polished using a Target Sample Preparation Unit (Leica TXP) and sectioned using Leica ultra-microtome (UC7) with a diamond knife at low speed to get 60nm-75 nm thick sections. Ultrathin sections were mounted on 200 mesh copper grids for imaging.

## Transmission electron microscopy imaging and analysis

High-resolution transmission electron microscopy (HRTEM) images were obtained using a Talos F200X Scanning/Transmission Electron Microscope equipped with a CETA 16 M camera at an accelerating voltage of 200 kV having a lattice-fringe resolution of 0.14 nm. All the relevant areas were marked using bright field (BF) imaging mode at spot size 5 and later scanned using the BF mode at spot size 5 and screen current between 3-4 nA. The data was analyzed using FIJI ImageJ software. For each sample, at least 80 individual mitochondria with clearly visible outer membrane were used for the analysis. A segmented line tool was used for manual tracking of mitochondrial parameters measurement. The mitochondrial perimeter /cristae length ratio was used to define the mass of the cristae and the mitochondrial length/width aspect ratio to define the circularity of mitochondria.

## Intracellular calcium measurement

Intracellular calcium was measured by using Indo-1 AM Calcium Sensor Dye (ThermoFisher Scientific). The protocol was similar to the previous measurement of mitochondrial mass, just cells were stained separately with Hoechst (4 μM) (Ex/Em 361/497) and Indo-1 AM (1 μM) (Ex/Em 346/475) due to similar excitation/emission spectra. Hoechst staining in control wells was used for checking seeding density between the samples and for normalization of Indo-1 AM dye in experimental wells. The visualized data represent the compilation of 3 independent experiments with 6 wells separately measured for each condition in each experiment.

## Metabolites measurement

For metabolites measurements, L-Lactate Assay Kit (ab65330), Pyruvate Assay Kit (ab65342), ADP assay kit (ab83359), ATP Assay Kit (ab83355), and Hydrogen Peroxide Assay Kit (ab102500) were purchased from Abcam and performed according to manufacturer's protocols for the assays. In short, NM1 WT, KO, and KO + NM1 MEFs were grown overnight in 6 well plates (300,000 cells/well), after which they were trypsinized, washed in PBS, and resuspended in the appropriate assay buffer. The protein concentration of each sample was quantified using the Pierce BCA protein assay kit (ThermoFisher Scientific). The amount of each metabolite was measured fluorometrically according to the manufacturer's recommendations and normalized to the total protein concentration of each sample. Each experiment was repeated at least 3 times with 6 replicates per sample per assay and final graphs are compiled from all experiments together.

## Glucose uptake assay

MEFs were grown overnight in 96 well plates (20000 cells/well) in full DMEM medium containing glucose (4).5 g/l), after which they were washed twice with 1xPBS and incubated in DMEM medium without glucose for 2 hr, 37 °C, 5%CO2. The medium was exchanged with fresh medium containing Hoechst stain (4 μM) and 2-NBDG fluorescent glucose analog (100 μg/ml) (ThermoFisher Scientific) and cells were incubated for an additional 30 min. The fluorescent intensity was measured for both dyes and relative glucose uptake normalized to Hoechst staining. The experiment was repeated 3 times with 6 replicates per sample and the final graph was compiled from all experiments together.

## Intracellular pH measurement

Intracellular pH measurement was performed with pHrodo Green AM Intracellular pH Indicator (ThermoFisher Scientific) according to the manufacturer's protocol. Cells were grown overnight in 96 well plates (20000 cells/well) in full DMEM medium followed by 30 min of incubation with Hoechst stain (4 μM) and pHrodo green dye mixed with PowerLoad concentrate in phenol red-free DMEM medium. After that, cells were washed twice with 1xPBS, and fluorescence intensity was measured for both dyes in a phenol red-free medium. The higher the pHrodo green fluorescent signal, the more acidic intracellular pH is. The pHrodo fluorescence data were normalized to the Hoechst stain. The experiment was repeated 3 times with 6 replicates per sample and the final graph was compiled from all experiments together.

## Large-scale ultra-performance liquid chromatography high-resolution mass spectrometry

5 replicates of NM1 WT and KO cells were used for metabolomic profiling. Overnight grown cells (-3 × 10^5 per sample) were washed twice with ice-cold 0.9% NaCl solution and 300 μl of 100% Methanol was added to each well. After 3 min of incubation on ice, cells were scraped using a precooled cell scraper and moved to a cold Eppendorf tube. Cell extracts were spun down at 15000 rpm for 15 min at 4 °C and 200 μl of each supernatant was moved to a new tube for further processing. Next, samples were vacuum dried and reconstituted in 200 μl

of cyclohexane/water (1:1) solution for subsequent analysis by Ultra Performance Liquid Chromatography High-Resolution Mass Spectrometry (UPLC-HRMS) performed at the VIB Metabolomics core facility (Belgium). In all, 10 μl of each sample were injected on a Waters Acquity UHPLC device connected to a Vion HDMS Q-TOF mass spectrometer. Chromatographic separation was carried out on an ACQUITY UPLC BEH C18 (50 × 2.1 mm, 1.7 μm) column from Watersunder the constant temperature of 40 °C. A gradient of two buffers was used for separation: buffer A (99:1:0.1 water:acetonitrile:formic acid, pH 3) and buffer B (99:1:0.1 acetonitrile:water: formic acid, pH 3), as follows: 99% A for 0.1 min decreased to 50% A in 5 min, decreased to 30% from 5 to 7 min, and decreased to 0% from 7 to 10 min. The flow rate was set to 0. 5 mL min$^{-1}$. Both positive and negative Electrospray Ionization (ESI) were applied to screen for a broad array of chemical classes of metabolites present in the samples. The LockSpray ion source was operated in positive/negative electrospray ionization mode under the following specific conditions: capillary voltage, 2.5 kV; reference capillary voltage, 2.5 kV; source temperature, 120 °C; desolvation gas temperature, 600 °C; desolvation gas flow, 1000 L h − 1; and cone gas flow, 50 L h⁻¹. The collision energy for the full MS scan was set at 6 eV for low energy settings, for high energy settings (HDMSe) it was ramped from 28 to 70 eV. The mass range was set from 50 to 1000 Da, scan time was set at 0.1 s. Nitrogen (greater than 99.5%) was employed as desolvation and cone gas. Leucine-enkephalin (250 pg/μL solubilized in water:acetonitrile 1:1 [v/v], with 0.1% formic acid) was used for the lock mass calibration, with scanning every 1 min at a scan time of 0.1 s. Profile data was recorded through Unifi Workstation v2.0 (Waters).

Data normalization was performed to remove potential variation resulting from instrument inter-run tuning differences. Raw MS peak data representing the abundance of each detected compound were subject to median standardization and missing values were imputed by the minimum value. Compounds with missing values in more than 50% of samples were considered missing. Standardized data that passed the quality control step were then log2 transformed and IQR normalized using JMP Genomics v8 (SAS Institute, Cary, NC) to remove potential technical artifacts and outliers. PCA and hierarchical clustering were done to explore the correlation structure in the data across the two conditions (WT and KO).

### Functional and metabolic pathway enrichment analysis of large-scale ultra-performance high-resolution LC-MS

Functional analysis of curated normalized peak data was performed using MetaboAnalyst v5.0 using an existing protocol[97]. Implemented Gene Set Enrichment Analysis (GSEA) method in the Functional Analysis module of MetaboAnalyst v5.0 (Accessed in 2022 from http://www.metaboanalyst.ca/) was used to identify sets of functionally related compounds and evaluate their enrichment of potential functions defined by metabolic pathways. GSEA analysis of compounds identified using positive and negative ionization was performed separately. Putative annotation of MS peaks data considering different adducts and ion modes was performed. *m/z* values and retention time dimensions both were used to increase confidence in identifying compounds and improve the accuracy of functional interpretations. Annotated compounds were then mapped onto *Mus musculus* (mouse) [KEGG][98] and a curated 912 metabolic data sets predicted to change due to dysfunctional enzymes based on human metabolism, separately[97], for pathway activity prediction (Supplementary Data 1). GSEA calculates the Enrichment score (ES) by walking down a ranked list of metabolites, increasing a running-sum statistic when a metabolite is in the metabolite set and decreasing it when it is not. A metabolite set is defined in this context as a group of metabolites with collective biological functions or common behaviors, regulations, or structures. In this method, the ES of each enriched pathway is calculated to reflect the degree to which a metabolite set is overrepresented at the top or bottom of a ranked list of metabolites. Each ES is then normalized by the average of all ES scores against all permutations of the expression dataset to generate normalized enrichment scores (NES) that are used to compare analysis results across metabolite sets. By normalizing the enrichment score, GSEA accounts for differences in metabolite set size and correlations between metabolite sets and the expression dataset. A positive NES indicates metabolite set enrichment at the top of the ranked list; a negative NES indicates metabolite set enrichment at the bottom of the ranked list[99]. Finally, compound hits were identified for each enriched pathway. Raw metabolomic data are publicly available in the Mendeley database (https://doi.org/10.17632/nxzs4dtztg.1; https://data.mendeley.com/datasets/nxzs4dtztg/2).

### Small-scale liquid chromatography mass spectrometry

A Blank (*n* = 3) and the WT, KO, and KO + NM1 extracted samples (randomized) were analyzed in a single analytical run using simultaneous ESI positive and negative ion modes with LC-MS. Quality control (QC) sample was prepared by pooling equal volumes of 20 μL from each sample in the study, vortexed for 30 s, and centrifuged at 13,000 g, at 4 °C for 5 min. Six injections of the pooled QC sample were analyzed at the beginning of the sample run to equilibrate the column before the analysis of the samples. Pooled QC injections were interspaced throughout the run to check the stability, robustness, repeatability, and performance of the analytical system. QC samples were also used for the identification of metabolites in the study. Chromatography was performed using the Vanquish UHPLC system (Thermo Fisher Scientific, Waltham, MA, USA) on a Kinetex HILIC column (2.1 × 100 mm, 2.6 μm particle size, 100 Å, Phenomenex, Torrance, CA, USA). The column was maintained at 30 °C and a flow rate of 200 μL/min. Mobile phases used were: (A) 0.1% formic acid, 10 mM ammonium formate in water, and (B) 0.1% formic acid in acetonitrile. The gradient started at 5% (A) and increased to 60% (A) over 10.5 min and held at 60% (A) for another 4.5 min. The composition was returned to its initial conditions of 5% (A) in 2 min. The flow rate was then increased to 300 μL/min in 1 min and the column was left to re-equilibrate for 12.5 min before returning to its initial flow rate of 200 L/min in 2 min (32 min total time). The injection volume was 5 μL and samples were maintained at 4 °C during the analysis. Orbitrap Fusion Lumos Tribrid Mass Spectrometer (Thermo Fisher Scientific, Waltham, MA, USA) was used in simultaneous ESI+ and ESI- modes for full LC-MS profiling and to generate data-dependent MS/MS (ddMS/MS) accurate mass spectra for identification (most intense peaks, TopN). The operational parameters were: spray voltage 3.5 kV (ESI + ), 3.5 kV (ESI-), sheath, auxiliary, and sweep gas flow rates were: 45, 8, and 1 (arbitrary units), respectively, for both modes. Capillary and heater temperatures were maintained at 350 °C and 300 °C, respectively. Data were acquired in full scan mode with a resolution of 60,000 from m/z 50–1000. TopN ddMS/MS scans were performed on the QC sample at a resolution of 17,500 and stepped collision energies (CE) of 20, 35, and 50.

The raw LC−MS dataset of the samples in the study was initially processed with Compound Discoverer 3.3 (Thermo Fisher Scientific, Waltham, MA, USA) for untargeted peak-picking, peak alignment, peak deconvolution and annotation of related peaks using a tailored untargeted metabolomics workflow. The generated raw LC-MS peaks were subjected to a QC correction approach in which the detected peaks must be present in ≥ 30% of the QC sample to be considered sample-related peaks for further analysis. The dataset was then normalized to a constant median and Log transformed to restore normality with 891 metabolic features (mz/rt) detected in all samples.

### Functional analysis of small-scale LC-MS

Data consisting of 891 metabolic features (mz/rt) were detected in 39 samples (11 WT, 13 KO, and 15 KO + WT). The peak data for the metabolic features were normalized by the sample median, log-transformed (base 10), and scaled using Pareto scaling (mean-centered

and divided by the square root of the standard deviation of each variable) using MetaboAnalyst v5.0 using an existing protocol[97]. Implemented mummichog pathway Enrichment Analysis method in the Functional Analysis module of MetaboAnalyst v5.0 (Accessed in 2023 from http://www.metaboanalyst.ca/) was used to identify sets of functionally related compounds and evaluate their enrichment of potential functions that are potentially rescued in the experiment. The analysis leverages the power of known metabolic pathways to gain functional insight directly from m/z features and retention times. The mummichog algorithm includes: 1) Permutations: A list of metabolites (the same length as the number of significant m/z features) are inferred from the set of m/z features, considering all potential matches (isotopes/adducts). Raw metabolomic data are publicly available in the Mendeley database (https://doi.org/10.17632/nxzs4dtztg.1; https://data.mendeley.com/datasets/nxzs4dtztg/2).

## HiC-Seq sample preparation and analysis

NM1 WT and KO cells were grown under standard conditions to 80% confluency with two biological replicates used for each condition. After trypsinization, cells were washed twice in 1x PBS buffer and 1 million cells per condition were fixed with 2% formaldehyde for 10 mins. The cell pellets were washed twice by 1× PBS, fast-frozen on dry ice, and stored at −80 °C. All subsequent processing and Hi-C were performed by the Genome Technology Center at NYU Langone Health, NY, using standard DNA extraction and library preparation protocols and the Arima Hi-C kit (Arima Genomics, San Diego, CA), respectively. Subsequently, sequencing libraries were prepared by using a modified version of the KAPA HyperPrep library kit (KAPA Biosystems, Willmington, MA) and sequenced by a NovaSeq instrument. Raw sequencing data were processed using the HiCUP[100] pipeline and analyzed using HOMER[101] (http://homer.ucsd.edu/homer/). For preprocessing with HiCUP, a digest file compatible with the Arima protocol was produced with the HiCUP digester using the option –arima. Processed bam files produced by HiCUP were converted to HOMER format using the script hicup2homer followed by conversion to homer tag directories using the command makeTagDirectory -format HiCsummary. PCA analysis was performed using HOMER with the command runHiCpca.pl -genome mm10 -res 500000 -window 500000 followed by annotation and differential analysis with the scripts annotatePeaks.pl and getDiffExpression.pl. Bins changing PC1 values from positive to negative or vice versa with an FDR of less than 0.05 between WT and KO cells were classified as switching. Protein-coding genes with TSS overlapping each compartment were identified using the value package[102]. TADs were called on replicate-merged tag directories of WT-MEFs using the HOMER function findTADsAndLoops.pl with -window and -res set to 50000. The insulation score of the identified domains in all replicates of each condition was extracted using the command findTADsAndLoops.pl with the "score" option followed by differential analysis using getDiffExpression.pl. Domains showing the change in insulation score with an adjusted p-value less than 0.05 were classified as a differential. Data is available in the Gene Expression Omnibus (GEO) database under accession number GSE198989 (https://www.ncbi.nlm.nih.gov/geo/query/acc.cgi?acc=GSE198989).

## ATAC-Seq sample preparation and analysis

NM1 WT and KO cells were used for the analysis performed commercially by Novogene (Beijing, China) with two biological replicates used for each condition. Isolated cell nuclei were mixed with Tn5 Transposase with two adapters for 30 min at 37 °C for DNA fragmentation, followed by DNA purification and amplification with a limited PCR cycle using index primers. Libraries were prepared according to recommended Illumina protocols and sequenced by NovaSeq 6000 instrument. ATAC-Seq processing and quality control were performed by Novogene (Beijing, China). Adapter trimming on raw fastq files was performed using trim-galore with default settings. Surviving paired

reads were aligned against the relevant reference genome (GRCm38) using Burrows-Wheeler Aligner BWA-MEM. The resulting BAM alignments were cleaned, sorted, and deduplicated (PCR and Optical duplicates) with PICARD tools (http://broadinstitute.github.io/picard). Processed bam files were converted to HOMER tag directories followed by annotation and differential analysis with the scripts annotatePeaks.pl and getDiffExpression.pl. ATAC-Seq peaks were called on cleaned, deduplicated bam files of both replicates of each condition together using macs2 with the parameters -q 0.05 -g mm/hg --keep-dup all --nomodel --shift −100 --extsize 200 -B --broad -f BAMPE. Peaks of the two conditions being compared were merged using the homer command mergePeaks and annotated with annotatePeaks.pl. Differential peaks were identified using the standard DESeq2 pipeline with lfcThreshold =1.5 and alpha=0.05. Data is available in the Gene Expression Omnibus (GEO) database under accession number GSE198988 (https://www.ncbi.nlm.nih.gov/geo/query/acc.cgi?acc=GSE198988).

## ChIP-Seq sample preparation and analysis

Cells were crosslinked (two biological replicates per condition and input controls) using 1% formaldehyde for 10 min followed by quenching with 0.125 M Glycine for 5 min and lysis with lysis buffer 1 (50 mM Hepes KOH pH 7.5, 10 mM NaCl, 1 mM EDTA, 10% glycerol, 0.5% NP-40, 0.25% Triton X-100). Nuclei were pelleted, collected, and washed using lysis buffer 2 (10 mM Tris-HCl pH 8, 200 mM NaCl, 1 mM EDTA, 0.5 mM EGTA). This was followed by lysis using lysis buffer 3 (10 mM Tris-HCl pH 8; 100 mM NaCl, 1 mM EDTA; 0.5 mM EGTA; 0.1% Na-Deoxycholate, 0.5% N-laurylsarcosine). Chromatin was sheared using Qsonica Sonicator (4 cycles of 3 min at 70% Amplitude) and then checked on 0.8% agarose gel. 100 μg of fragmented chromatin was mixed with the appropriate antibody. The protein-antibody immuno-complexes were recovered by the Pierce Protein A/G Magnetic Beads. Beads and attached immunocomplexes were washed twice using Low salt wash buffer (0.1% SDS; 2 mM EDTA, 1% Triton X-100, 20 mM Tris-HCl pH 8, 150 mM NaCl), and High Salt wash buffer (0.1% SDS, 2 mM EDTA, 1% Triton X-100, 20 mM Tris-HCl pH 8, 500 mM NaCl), respectively. The beads were resuspended in elution buffer (50 mM Tris-HCl pH 8, 10 mM EDTA, 1% SDS). De-crosslinking was achieved by adding 8 μL of 5 M NaCl and incubating at 65 °C overnight. RNase A (1 μL 10 mg/mL) was added for a 30 min incubation at 37 °C. Then, 4 μL 0.5 M EDTA, 8 μL 1 M Tris-HCl, and 1 μL 20 mg/mL proteinase K (0.2 mg/mL) were added for a 2-h incubation at 42 °C to digest the chromatin. DNA was purified by a QIAquick PCR purification kit for qPCR analysis and sequencing. Raw reads were quality trimmed using Trimmomatic[103] and analyzed with FastQC (http://www.bioinformatics.babraham.ac.uk/projects/fastqc) to trim low-quality bases, systematic base-calling errors, and sequencing adapter contamination. Specific parameters used were "trimmomatic_adapter.fa:2:30:10 TRAILING:3 LEADING:3 SLIDINGWINDOW:4:15 MINLEN:36". Surviving paired reads were then aligned against the mouse reference genome (GRCm38) using Burrows-Wheeler Aligner BWA-MEM. The resulting BAM alignments were cleaned, sorted, and deduplicated (PCR and Optical duplicates) with PICARD tools (http://broadinstitute.github.io/picard). Bigwig files were generated using deeptools[104] command bamCoverage -bs 10 -e --ignoreDuplicates –normalizeUsingRPKM after excluding encoded blacklisted regions. Bigwig files were analyzed with the computeMatrix function of deeptools to plot the average signal around regions of interest. Peaks were called using macs2[105] on replicate-merged bam files with relevant input control and q = 0.05. The –broad flag was used for H3K27me3 while narrow peaks were called for other epigenetic marks. Peaks of the two conditions being compared were merged using the homer command mergePeaks followed by annotation and differential analysis with the scripts annotatePeaks.pl and getDiffExpression.pl. Peaks showing differential expression with an adjusted p-value < 0.05 were classified as

differentially expressed. Data is available in the Gene Expression Omnibus (GEO) database under accession number GSE202716 (https://www.ncbi.nlm.nih.gov/geo/query/acc.cgi?acc=GSE202716).

## In vivo tumor formation studies

Female Balb/c nude mice (16 weeks old; 25–35 g in weight) were purchased from Jackson Laboratory, USA. The mice were housed in a pathogen-free sterile ventilation system supplied with sterile woodchip and ad libitum feed (PicoLab Rodent Diet 20, 5053) and water supply. Mice were kept in a room maintained at a temperature of 21–24 °C on a schedule of 12 h light/dark cycle. All animal experiments were performed after approval by the NYUAD-IACUC (Protocol 21-0005).

A total of 15 mice were divided into three groups of 5 and injected with either WT, KO, or KO + NM1 embryonic fibroblasts (0.1 mL, $3 \times 10^6$ cells/100 μL in 10 mM PBS, pH 7.4) subcutaneously into the right flank. Mice were weighed daily and monitored for mobility, respiratory distress, and signs of pain for up to 28 days. Tumor size was measured daily using high-precision calipers. As per the humane endpoint of the NYUAD-IACUC protocol approvals, the maximum tumor permitted is 2000 mm3. In this study, the maximum tumor volume has not exceeded this limit. At the end of the experiment, all mice from each group were euthanized by $CO_2$ exposure and perfused with isotonic PBS followed by fixation with 10% neutral buffered formalin. Tumor and mammary pad tissues were rapidly removed, weighed, and kept in 10% neutral buffered formalin for subsequent analysis[106].

## Immunohistochemistry and histopathology

Expression of Mct1, Egfr, Bclxl, and Ki-67 tumorigenic factors was probed by immunohistochemistry (IHC). The excised and fixed tumor and mammary pad tissues were paraffin-embedded, cut into 4 μm thick sections with a microtome (Leica microtome) and mounted on glass slides, deparaffinized by incubation in 3% methanol-hydrogen peroxide solution, followed by treatment with 10 mM EDTA (pH 8.0) at 95 °C. Thereafter, the slides were cooled to room temperature, rinsed with a phosphate-buffered saline solution containing 0.05% of tween 20 (PBST), and incubated with individual primary antibodies (Egfr, Ki-67, E-cadherin, Bcl-xl, Mct1) for 2 h. Subsequently, the slides were rinsed with PBST and incubated with the appropriate HRP-conjugated secondary antibody at room temperature for 30 min, counterstained with hematoxylin, dehydrated in ethanol, and mounted with DPX mounting media for further microscopic analysis. For the histopathological analysis, tissues from the lungs, liver, spleen, kidney, and heart, along with mammary pad and tumor tissues, were fixed with neutral buffered formalin, paraffin-embedded, and sectioned for hematoxylin and eosin (H & E) staining. The tissue sections were then imaged using light microscopy[107].

## RNA-Seq library preparation, sequencing, and analysis

For transcriptional profiling of tumor tissues, tumors derived from KO cells and tumors derived from KO + NM1 cells were selected for comparison. Each tumor was divided into equal pieces and each piece was separately homogenized in RNAzol reagent by using the Bead Ruptor 96 Homogenizer (Omni International). The RNA-Seq library was prepared by using the NEBNext Ultra II RNA Library Prep Kit for Illumina (NEB) and sequenced with the NextSeq 500/550 sequencing platform (performed at the NYUAD Sequencing Center). All of the subsequent analysis, including quality trimming, was executed using the BioSAILs workflow execution system. Trimmomatic (version 0.36) was used for quality trimming of the raw reads to get rid of low-quality bases, systematic base-calling errors, as well sequencing adapter contamination[103]. The quality of the sequenced reads pre/post quality trimming was assessed by FastQC and only the reads that passed quality trimming in pairs were retained for downstream analysis (https://www.bioinformatics.babraham.ac.uk/projects/fastqc/). The quality-trimmed RNA-Seq reads were aligned to the *Mus musculus* GRCm38 (mm10)

genome using HISAT2 (version 2.0.4)[108]. The conversion and sorting of SAM alignment files for each sequenced sample to BAM format were done by using SAMtools (version 0.1.19)[109]. The BAM alignment files were processed using HTseq-count, using the reference annotation file to produce raw counts for each sample. The raw counts were then analyzed using the online analysis portal NASQAR (http://nasqar.abudhabi.nyu.edu/), to merge, normalize, and identify differentially expressed genes (DEG). DEG by at least twofold ($\log_2(FC) \geq 1$ and adjusted p-value of <0.05 for upregulated genes, and $\log_2(FC) \leq -1$ and adjusted p-value of <0.05 for downregulated genes) between the tumors derived from KO and KO + NM1 were subjected to GO enrichment using DAVID Bioinformatics (https://david.ncifcrf.gov/)[110]. RNA-Seq data were deposited in the Gene Expression Omnibus (GEO) database under accession number GSE236679 (https://www.ncbi.nlm.nih.gov/geo/query/acc.cgi?acc=GSE236679).

The RNA-Seq data set of NM1 WT and KO primary mouse embryonic fibroblast used in this study (Fig. 1d, e, f), was described previously[23] and is available in the Gene Expression Omnibus (GEO) database under accession number GSE133506 (https://www.ncbi.nlm.nih.gov/geo/query/acc.cgi?acc=GSE133506). The parameters and subsequent analysis used to define differentially expressed genes were the same as for RNA-seq analysis of tumor tissues. The list of genes used for comparative analysis was based on GO terms "oxidative phosphorylation" (GO:0006119) and "mitochondrion" (GO:0005739) (http://geneontology.org)[111,112].

## COSMIC database

Mutagenesis rate and expression status comparisons of *Myo1C*, *p53*, and *mTOR* in human cancer samples are based on the data downloaded from COSMIC - Catalogue of Somatic Mutations in Cancer (version COSMIC v96)[62].

## TCGA RNAseq data

All analyses on the TCGA data were performed using R (v 4.2.1). The RNAseq STAR counts data were downloaded and processed using Biolinks (v 2.24.3)[113]. The RNAseq STAR counts data for 32 solid cancer types were downloaded from the GDC portal (https://portal.gdc.cancer.gov/repository) and processed using Biolinks (v 2.24.3)[113]. TCGA primary tumor samples were extracted using TCGA-Assembler 2 (v 2.0.3)[114]. For skin cutaneous melanoma (SKCM), the primary tumor sample is frequently unavailable for genomic analysis and TCGA mostly includes lymph node metastatic samples, which were also included in the analysis. In the case of duplicate aliquots/samples, one was removed randomly. Only one sample for the patient was included. Gene symbols were converted to official HGNC gene symbols. Gene expression normalization was performed within lanes, to correct for gene-specific effects (including GC-content) and between lanes, to correct for sample-related differences (including sequencing depth) using EDASeq (Exploratory Data Analysis and Normalization for RNA-Seq) (v 2.12.0)[115]. The resulting expression values were quantile normalized using preprocessCore (v 1.36.0)[116].

## TCGA somatic mutation data

Somatic mutation calls from the TCGA MC3 Project were downloaded using TCGAmutations (https://github.com/PoisonAlien/TCGAmutations)[117]. The MAF file was filtered to only include non-synonymous mutations ("Frame_Shift_Del", "Frame_Shift_Ins", "In_Frame_Del", "In_Frame_Ins", "Missense_Mutation", "Nonsense_Mutation", "Splice_Site", "Translation_Start_Site", "Nonstop_Mutation"). Samples were stratified into two groups: with and without non-synonymous mutations in the *Myo1C* gene. Tumor samples with both mutations and RNAseq data available were used for the analysis. These include samples from the following tumor types: uterine corpus endometrial carcinoma (UCEC; $n = 528$), colon adenocarcinoma (COAD; $n = 402$), skin cutaneous melanoma (SKCM; $n = 465$), stomach adenocarcinoma

 

(STAD; $n = 372$), head and neck squamous cell carcinoma (HNSC; $n = 495$), lung adenocarcinoma (LUAD; $n = 511$), breast invasive carcinoma (BRCA; $n = 1017$), bladder urothelial carcinoma (BLCA; $n = 405$), kidney renal papillary cell carcinoma (KIRP; $n = 280$), brain lower grade glioma (LGG; $n = 509$), liver hepatocellular carcinoma (LIHC; $n = 358$), lung squamous cell carcinoma (LUSC; $n = 482$), thyroid carcinoma (THCA; $n = 488$), cervical squamous cell carcinoma and endocervical adenocarcinoma (CESC; $n = 286$), esophageal carcinoma (ESCA; $n = 161$), ovarian serous cystadenocarcinoma (OV; $n = 257$), cholangiocarcinoma (CHOL; $n = 35$), glioblastoma multiforme (GBM; $n = 150$), kidney renal clear cell carcinoma (KIRC; $n = 366$), pancreatic adenocarcinoma (PAAD; $n = 168$), prostate adenocarcinoma (PRAD; $n = 493$), sarcoma (SARC; $n = 234$), uveal melanoma (UVM; $n = 80$), adrenocortical carcinoma (ACC; $n = 79$), lymphoid neoplasm diffuse large B-cell lymphoma (DLBC; $n = 37$), kidney chromophobe (KICH; $n = 65$), mesothelioma (MESO; $n = 82$), pheochromocytoma and paraganglioma (PCPG; $n = 178$), testicular germ cell tumors (TGCT; $n = 144$), thymoma (THYM; $n = 119$), uterine carcinosarcoma (UCS; $n = 57$), rectum adenocarcinoma (READ; $n = 145$). Altogether, there were 9448 samples available for the Pan-cancer analysis.

## TCGA oncogenic pathway enrichment score
Gene sets that reflect cancer-cell intrinsic (oncogenic) pathways were selected from multiple sources as described in detail in Roelands et al[118]. After removing redundant pathways, 43 oncogenic pathways were included in the final analysis. Single sample gene set enrichment analysis (ssGSEA) was applied to the log2 transformed, normalized gene expression matrix, using GSVA (v 1.44.5)[119], to generate the pathway enrichment score (ES) for each sample.

## TCGA regression analyses
Linear regression models were fit to define the association between Myo1C non-synonymous mutations and the oncogenic pathways' ES using the lm function. This analysis was performed in the entire dataset (Pan-cancer, $n = 9448$) as well as within each cancer type separately (Per-cancer). In the Pan-cancer analysis, we used the Benjamini-Hochberg FDR to correct for multiple hypothesis testing. Cancer types with less than three samples harboring Myo1C non-synonymous mutations (ACC, GBM, KIRC, KICH, MESO, SARC, DLBC, READ, PCPG, TGCT, CESC, THYM, ESCA, UCS, PAAD, PRAD, CHOL, UVM) were excluded from the Per-cancer analysis but retained in the Pan-cancer analysis. The results were visualized using ComplexHeatmap (v.2.1.2)[120] and ggplot2 (v.3.4.1)[121]. Associations with an FDR < 0.1 in the Pan-cancer analyses were plotted. Data used are extracted from an open-access repository (https://portal.gdc.cancer.gov/repository). All data relevant to the study are included in the article (Supplementary data 3–7)

## Statistics and reproducibility
GraphPad Prism 8.3.0 software was used for statistical analysis. An unpaired $t$-test was used in all the experiments unless otherwise stated in the text. Error bars in boxplots represent minimum and maximum values. Error bars in bar charts represent SD. Statistical significance is marked with $*p < 0.05$, $**p < 0.01$, $***p < 0.001$, $****p < 0.0001$, ns (not significant). Each experiment was performed at least in triplicates. The exact description of sample sizes and number of replicates is defined for each experimental procedure in Methods.

## Reporting summary
Further information on research design is available in the Nature Portfolio Reporting Summary linked to this article.

## Data availability
Source data for the majority of experiments are provided as a Source Data file. Tumor RNA-Seq data were deposited in the Gene Expression Omnibus (GEO) database under accession number GSE236679

[https://www.ncbi.nlm.nih.gov/geo/query/acc.cgi?acc=GSE236679]. The RNA-Seq data set of NM1 WT and KO primary mouse embryonic fibroblast used in this study (Fig. 1d–F), was described previously[23] and is available in the Gene Expression Omnibus (GEO) database under accession number GSE133506. Hi-C data is available in the Gene Expression Omnibus (GEO) database under accession number GSE198989. ATAC-Seq data is available in the Gene Expression Omnibus (GEO) database under accession number GSE198988. ChIP-Seq data is available in the Gene Expression Omnibus (GEO) database under accession number GSE202716. Raw metabolomic data from large-scale metabolomic analysis are publicly available in the Mendeley database (https://doi.org/10.17632/nxzs4dtztg.1; https://data.mendeley.com/datasets/nxzs4dtztg/2). Raw metabolomic data from small-scale metabolomic analysis are publicly available in the Mendeley database (https://doi.org/10.17632/nxzs4dtztg.1; https://data.mendeley.com/datasets/nxzs4dtztg/2). Source data are provided with this paper.

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

## Acknowledgements

This work is supported by grants from NYU Abu Dhabi, the Sheikh Hamdan Bin Rashid Al Maktoum Award for Medical Sciences, Cancerfonden (Swedish Cancer Society), a donation from the Cipriani family, and Tamkeen under the NYU Abu Dhabi Research Institute Award to the NYUAD Center for Genomics and Systems Biology (ADHPG-CGSB) to P.P. We thank the NYU Abu Dhabi Center for Genomics and Systems Biology, in particular Marc Arnoux and Mehar Sultana for RNA sequencing, as well as Core Technology Platform Resources, including the NYU Abu Dhabi imaging center, in particular Rachid Rezgui and Rainer Straubinger for their help with the microscopy. We appreciate the computational platform provided by the Center for Genomics and Systems Biology and the NYU Abu Dhabi HPC team and are especially thankful to Nizar Drou for technical help.

## Author contributions

T.V. and P.P. conceived the research and wrote the manuscript. T.V. performed all RNA-seq analyses, imaging, biochemical experiments, and data analysis and prepared the figures. O.S. performed high-content screening and helped with the biochemical phenotyping of NM1 KO cells. W.S.A., Sa.A., Y.I., and Sh.A. performed metabolomics analysis. N.H.E. and S.R.M. performed ChIP-seq, ATAC-seq, and HiC-seq analysis. S.T. and R.P. performed electron microscopy experiments, M.M. and P.L. planned and performed the NM1 KO tumor studies in nude mice, and S.S. and D.B. performed correlation analysis of Myo1C mutations in human cancers. P.P. supervised the research. All authors read and approved the manuscript.

## Competing interests

The authors declare no competing interests.
