## [Peer Review File · Nature Communications]

Positive regulation of oxidative phosphorylation by nuclear myosin 1 protects cells from metabolic reprogramming and tumorigenesis in miceREVIEWER COMMENTS

Reviewer #1 (Remarks to the Author):

The manuscript entitled “Regulation of oxidative phosphorylation by nuclear myosin 1 protects cells from metabolic reprogramming and tumorigenesis in mice” is describing an excellent study, well designed and experimentally sound. The study gives novel insights into how cells respond to environmental cues, and demonstrates that NM1 is a key component. The finding that NM1 deletion directly causes tumours is of major importance and is thoroughly tested. The underlying cause is investigated and points to a metabolic shift and a dysregulation of mitochondria caused by a deregulated mTORC1 signalling. This part, even if the experiments are solid, should be further investigated to back the conclusions.

Figure 1 – All transcript levels tested in Figure 1C and D display a downregulation whereas the NM1 KO sequencing data shows upregulation of some mitochondrial associated RNAs. (Even the *Uqcrb* is upregulated in the sequencing data and down in the qPCR.) Are the transcripts that are found at a higher level regulated differently than the other mitochondrial genes?

Some of the genes differently expressed are normally expressed in a cell type specific manner. Does NM1 knock down change the gene expression pattern in MEFs?

Figure 2 – E) A further mitochondrial protein, such as TOM20, should be used to detect mitochondria in the images, since the expression of TFAM is also affected in NM1 KO MEFs.

Figure 3 – B) The transcriptional effect if Pink1 would suggest that no protein is present in the cell and cannot be stabilised upon mitochondrial dysfunction. Is this seen in NM1 KO cells, with a lack of Parkin activation for instance? In addition, *Snca*-RNA is also done. What is the functional consequence of this?

Are the PINK1 and SNCA genes direct targets of NM1? These genes should also be investigated for NM1 binding, similar to the transcription factors in Figure 4, and for chromatin accessibility.

Figure 4 – C and F) Is the promoter architecture similar of the genes investigated, or the factor requirement? The histone modifications tested seem to correlate with NM1 binding, except for H3K4me3 in PGC1a. It has been shown that NM1 recruits PCAF. Is PCAF involved in the transcriptional regulation of all genes? Which histone methyl transferase is involved? Could these factors be affected by NM1 deletion?

G) The fraction phosphorylated seems to be minor so it would be of interest to know which specific position is phosphorylated by mTORC1. And if this site is important for recruitment to the genes.

Supplementary Figure 2: Is the lack of chromatin changes at the promoter upon NM1 deletion because these genes are already in an open configuration, maybe with a paused RNA polymerase II?

Figure 6 – H –J: The heat maps should be better explained, what is compared to what?

What is the reason for the differential expression of MYO1C in slow vs fast growing? A qPCR of NM1 should be performed to more directly assess the expression level, as well as estimating the protein level and the phosphorylation level of NM1. If possible, the recruitment of NM1 to the gene promoters would also provide greater insights into the differences in slow vs fast growing tumours.

Have further mutations/deletions been acquired in the fast vs slow growing tumours? Is p53 mutated/activated in the fast growing cells?

Discussion – The result about mitochondrial structural maintenance is not addressed, and should be further developed, preferentially with more experimental data.

The discussion about tumour formation should have more experimental backing, see comment Figure 6.

Reviewer #2 (Remarks to the Author):

Tomas Venit and colleagues study the function of the nuclear myosin 1 (NM1) protein. NM1 is a chromatin remodeling protein. Using a CRISPR-Cas9-mediated knockout cell clone of mouse embryonic fibroblasts (KO-MEFs), the authors perform a long series of detailed and well-controlled experiments, focusing primarily on the function, homeostasis and gene-mediated integrity of mitochondria and their metabolism. Loss of NM1 from these normal MEFs leads to Warburg-like effect and oncogenic transformation of the fibroblasts in the absence of cooperating oncogenes or loss of tumor suppressor genes. This established NM1 as a tumor suppressor. These interesting results are presented in Figs 1-3 and. In Fig. 4 the paper tries to generate a mechanism of action of NM1 and presents data that potentially link NM1 to the oncogenic mTOR pathway. mTOR possibly phosphorylates NM1, and NM1 binds to genes that express transcription factors that regulate mitochondrial gene expression. NM1 is also shown to regulate mTOR signaling in a positive feedback mechanism. Finally, transcriptomic analyses of tumors generated by the NM1-KO MEFs are presented in Fig. 6 that present again misregulation of mitochondrial genes.

The paper presents a series of well-controlled experiments and combines high-throughput analyses to more focused analyses of specific molecular pathways. This reviewer has two major conceptual issues with the presented data: a) the potency of NM1 KO leading to tumorigenesis in mice is very high. One therefore would expect a similar function of NM1 in human tumors. Accordingly, the paper presents the data of Fig. 6L and 6M, which are very important but correlative. No evidence for an equivalent mechanism in human tumors is presented. b) It is unclear which tumor suppressor signaling pathway mediates NM1 phosphorylation via mTOR, which then maintains mitochondrial homeostasis by regulating transcription of PGC1a and TFAM. This core question remains unanswered and characterizes the essential message of this paper.

I here present some additional comments on specific results of this paper.

1. Title: "Regulation of OXPHOS..." is rather vague. It would be nice if the title indicated more accurately the proposed mechanism: positive regulation of genes that promote mitochondrial biogenesis.
2. The RNAseq data of Fig. 1E are difficult to understand. The legend indicates that the genes are listed in descending order. What does this mean? Most genes show the same expression pattern in WT and a different pattern in KO MEFs. The genes at the bottom of the lists show the inverse pattern. This is not explained. Why is this for only a few genes in each of the 5 complexes?
3. The data of Suppl. Fig. 2C and 2D are important but there are not explained to a non-specialist. What is A and B and what do the changes supposedly represent? Based on these data and the ATACseq data the authors conclude in p.10, lines 292-294 that NM1 deletion does not affect the expression of mitochondrial genes directly by changing the chromatin landscape or global 3D genome architecture, ... through direct regulation of ...transcription factors. The terms directly and direct generate confusion. Which protein directly changes genome architecture? The histones? If NM1 acts as a chromatin remodeling factor, how are the ATACseq and HiC data interpreted? Is the impact of NM1 to the specific transcription factors of mitochondrial genes dependent on the ability of NM1 to regulate chromatin remodeling?
4. Related to the above comment, does the CHIP-qPCR analysis of Fig. 4 agree with the ATACseq analysis of the PGC1a, TFAM and mTOR genes?
5. The data of Fig. 4G on NM1 phosphorylation are very interesting. Phospho-peptide motif analysis for mTOR targets is worth performing and verification of the phosphorylation by immunoprecipitation-western blot analysis.
6. P.11, lines 327-329: NM1 functions as part of a nutrient-sensing signaling pathway. The evidence that supports this evidence should be clearly presented.
7. The potency of NM1 KO in generating tumors in mice requires explanation.
8. As commented above, the paper contends that nutrient sensing is at play upstream of NM1 regulation. Yet Fig. 7 indicates growth factor signaling as acting upstream. Are growth factors implied by the term "nutrient-sensing"?

Reviewer #3 (Remarks to the Author):

This is a potentially interesting study but can not be meaningfully evaluated because it is completely missing an absolutely critical control - stable reconstitution (addback) of NM1 in NM1-knockout cells. Creating addback controls is important for any KO cell line but is especially important in clonal KO lines which these appear to be based on the referenced previous publication from this group (Venit et al Commun Biol 2020). These control cells are required to show that the NM1-KO phenotypes can be rescued by adding back NM1 and that the differences between cell lines did not arise due to off-target effects of CRISPR, unrelated genetic or epigenetic changes that arose during KO cell line establishment, or other random reasons. Even better, if a loss of function point mutant of NM1 can be created that

lacks activity, this should also be used to make an a control cell line in addition to WT NM1. Unfortunately, without this basic control it is pointless to evaluate the data in depth.

I would be happy to review an updated version of this manuscript that includes NM1 addback controls for most experiments.

REVIEWER COMMENTS

We would like to thank all reviewers for their constructive feedback. Please find below our point-by-point responses (black text) to each of the reviewers' concerns (red text).

Reviewer #1 (Remarks to the Author):

The manuscript entitled "Regulation of oxidative phosphorylation by nuclear myosin 1 protects cells from metabolic reprogramming and tumorigenesis in mice" is describing an excellent study, well designed and experimentally sound. The study gives novel insights into how cells respond to environmental cues, and demonstrates that NM1 is a key component. The finding that NM1 deletion directly causes tumours is of major importance and is thoroughly tested. The underlying cause is investigated and points to a metabolic shift and a dysregulation of mitochondria caused by a deregulated mTORC1 signalling. This part, even if the experiments are solid, should be further investigated to back the conclusions.

Figure 1 – All transcript levels tested in Figure 1C and D display a downregulation whereas the NM1 KO sequencing data shows upregulation of some mitochondrial associated RNAs. (Even the Uqcrb is upregulated in the sequencing data and down in the qPCR.) Are the transcripts that are found at a higher level regulated differently than the other mitochondrial genes? Some of the genes differently expressed are normally expressed in a cell type specific manner. Does NM1 knock down change the gene expression pattern in MEFs?

Thanks for bringing this up. We believe this is a slight misunderstanding of the data and we have done our best to clarify it in the revised manuscript. In figures 1C and D we performed RtgPCR and WB analyses on stable WT and NM1 KO cells which were prepared by CRISPR/Cas9 technology whereas figure 1E shows RNA-seq data from primary mouse embryonic fibroblasts isolated from WT and NM1 KO mouse embryos. We reported the results of this RNA-seq data in a recent study (please see Venit et al., 2020). Here we only plotted OXPHOS genes to show that regardless of the model system, NM1 deletion leads to similar phenotypic outcomes.

Therefore, even though there is a similar pattern of gene expression, some differences between the two experiments are likely to occur and could be simply explained by the use of different experimental models.

As mentioned above, all this is now explained in the main text of the revised manuscript where we discuss both the use of different experimental models and the observed differences.

Figure 2 – E) A further mitochondrial protein, such as TOM20, should be used to detect mitochondria in the images, since the expression of TFAM is also affected in NM1 KO MEFs.

The TFAM staining in the IF experiment wasn't meant to be used for any mitochondrial quantification. We have used only to show its reduced expression and redistribution within mitochondria. We believe that monitoring TOM20 or any other specific mitochondrial

protein may be misleading as two major mitochondrial factors responsible for expression of the majority of mitochondrial genes are deregulated in NM1 KO cells. For this reason, in the paper we used mitotracker dyes, ideally suited for mitochondrial staining and quantification in all experiments as they covalently bind the thiol groups of the cysteine residues of mitochondrial proteins (Chazotte, 2011). They can really serve as ideal control to study global differences in mitochondrial volume, structure and content.

We changed the organization of the figures and moved the data about DNA copy number in NM1 KO cells from figure 3 to figure 1 which helped us to better explain the reasoning behind the TFAM staining. In the revised draft, we also clarify the use of mitotrackers.

Figure 3 – B) The transcriptional effect if Pink1 would suggest that no protein is present in the cell and cannot be stabilised upon mitochondrial dysfunction. Is this seen in NM1 KO cells, with a lack of Parkin activation for instance? In addition, Snca-RNA is also down. What is the functional consequence of this?

The Parkin protein was not detectable in MEFs and even though we found Pink1 and SNCA to be downregulated in NM1 KO cells by RtgPCR their expression in MEFs is generally very low and they cannot be visualized by Western blot or IF. So, mouse embryonic fibroblasts may not be a suitable model system to study the interplay between Parkin, SNCA and Pink1. However, in an ongoing study focusing on the role of NM1 in neuronal differentiation and brain development, one of the plans is to study the aforementioned proteins and their relationship with NM1 as they are more abundant in brain tissues in comparison to MEFs. Initial evidence suggests that at least Pink1 and Parkin protein levels in brain tissues seem to be NM1-dependent (see figure below).

Are the PINK1 and SNCA genes direct targets of NM1? These genes should also be investigated for NM1 binding, similar to the transcription factors in Figure 4, and for chromatin accessibility.

This is a great point. A possible scenario suggests that NM1 may indeed regulate Pink1 and SNCA either directly or indirectly. Answering this question, however, is somewhat beyond the scope of the present paper and it is technically very challenging in the current model system as explained above. For this reason we are currently addressing this issue in a follow-up study.

Figure 4 – C and F) Is the promoter architecture similar of the genes investigated, or the factor

requirement? The histone modifications tested seem to correlate with NM1 binding, except for H3K4me3 in PGC1a. It has been shown that NM1 recruits PCAF. Is PCAF involved in the transcriptional regulation of all genes? Which histone methyl transferase is involved? Could these factors be affected by NM1 deletion?

NM1 has been shown to recruit not only PCAF but also Set1b which is the methyltransferase responsible for methylation of H3K4 (Sarshad et al. 2013, Almuzzaini et al. 2015, Venit et al. 2020). This reviewer is correct that NM1 deletion has a lesser effect on histone methylation at TSS of the PGC1 α gene in comparison to mTOR and TFAM, but the methylation is still significantly reduced in comparison to the WT condition. The difference in acetylation and methylation levels between different genes could be explained by different PCAF and Set1B association with different combinations of transcription factors. For example, PCAF is targeted to DNA not only by NM1 but also by the actin-hnRNPU complex (Obrdlik et al., 2008) and p53 (Liu et al., 1999). Similarly, Set1B has been recently shown to be associated with HIF 1 complex to activate glycolytic genes during hypoxia conditions (Ortmann et al., 2021). This is interesting because at least in colorectal cancer, PGC1a expression is upregulated by hypoxia and leads to increased tumorigenesis (Yun 2019). As NM1 deletion leads to induction of a glycolytic program, it is possible that loss of H3K4 methylation on PGC1a TSS is partially rescued by glycolytic activation of Set1B.

In the revised manuscript we have now modified both introduction and results to more accurately address this question.

G) The fraction phosphorylated seems to be minor so it would be of interest to know which specific position is phosphorylated by mTORC1. And if this site is important for recruitment to the genes.

Phospho-proteomic analysis of NM1 showed a specific phosphorylation site on the Serine amino acid residue in position 1020. This phosphorylation event is dependent on GSK3 β and it happens during the G1 phase. It is important because it promotes NM1 binding to the chromatin and at the same time it protects NM1 from proteasomal degradation (Sarshad et al., 2014). Currently we don't know whether mTORC1 phosphorylation of NM1 is direct or indirect and whether it occurs on the same site but here we speculate that mTOR could regulate NM1 indirectly via phosphorylation of GSK3 β as it was suggested by several other studies (Zhang et al. 2006, Evangelisti et al. 2020, Bautista et al. 2018). We discuss this in more detail in the revised manuscript.

Supplementary Figure 2: Is the lack of chromatin changes at the promoter upon NM1 deletion because these genes are already in an open configuration, maybe with a paused RNA polymerase II?

At this stage we can only speculate. It is possible that NM1 directly binds and specifically regulates a subset of genes such as TFAM, PGC1a and mTOR and that the broader effects on transcription are a domino effect resulting from dysregulation of smaller sets of genes. In any case, as part of a follow up project, it would definitely be exciting to explore whether NM1 depletion leads to a stalling of the RNA polymerase machinery at least on those genes that are directly impacted by NM1.

We have now included ATAC-seq and ChIP-seq profiles for SNF2H, H3K9ac and H3K4me3 along TFAM, PGC1a and mTOR genes to show NM1-dependent changes. Remarkably, the majority of marks in NM1 KO cells support our ChIPqPCR data. These data sets are now part of supplementary figure 3 and they are discussed in the text.

Figure 6 – H –J: The heat maps should be better explained, what is compared to what? What is the reason for the differential expression of MYO1C in slow vs fast growing? A qPCR of NM1 should be performed to more directly assess the expression level, as well as estimating the protein level and the phosphorylation level of NM1. If possible, the recruitment of NM1 to the gene promoters would also provide greater insights into the differences in slow vs fast growing tumours.

Thanks for raising this point. The expression profile of Myo1C found in RNA seq of fast vs slow growing tumors was intriguing so we tested its expression by RtgPCR. After a close look we discovered that Myo1C gene expression is independent of tumor size and therefore we discard the previous data from the paper. In the revised manuscript we focused more on Myo1C in the context of human tumorigenesis and found that mutations in Myo1C are associated with increased proliferation and apoptosis, decreased mTOR signaling and increased glycolysis. We believe that this nicely correlates with the phenotypes observed in our NM1 KO cells. The results of this analysis are included in the revised manuscript as part of figure 7.

NM1 is one of the three alternatively spliced isoforms derived from the Myo1C gene. To generate tumors, we inject immunosuppressed mice with our NM1-specific KO mouse embryonic fibroblasts. Therefore, as the tumors that are generated from these cells do not express NM1, it is not possible to perform qPCR analysis, estimate NM1 protein levels and its phosphorylation status in the tumor tissue.

The heatmaps are now simplified and better explained in the figure, figure legends and main text.

Have further mutations/deletions been acquired in the fast vs slow growing tumours? Is p53 mutated/activated in the fast-growing cells?

Thanks for these questions. We have previously shown that NM1 directly synergizes with p53 to regulate p21 gene expression upon DNA damage and NM1 KO cells lacking the p21-regulated checkpoint are unable to resolve the broken DNA and accumulate DNA damage over generations. Although we do not know the state of p53 (whether mutated or activated) in the tumors generated from the NM1 KO MEFs, p53 expression at both mRNA and protein levels as well as its phosphorylation and binding to the p21 gene does not seem to be affected in NM1 KO MEFs. So, it seems more likely that p53 is functional in the absence of NM1 but downstream signaling is affected by changes in p21 expression (see Venit et al, 2020).

Discussion – The result about mitochondrial structural maintenance is not addressed, and should be further developed, preferentially with more experimental data.

We agree that this is an important point. As our MEFs do not seem to be an ideal model to study mitochondrial dynamics and turnover, we have reprogrammed MEFs to neurons to

study mitochondrial fission and fusion and their movement along axons in a controlled manner. Preliminary unpublished results show that NM1 KO-induced neurons seem to have a problem to properly form axons and they are missing mitochondria in neuronal boutons which are places of neuronal synapses (see figure below). Although this phenotype is currently being under investigation, these preliminary results are compatible with a role of NM1 in mitochondrial biogenesis and dynamics. These results are part of an independent study that is geared to understand how NM1 may function in mitochondrial structural maintenance.

The discussion about tumour formation should have more experimental backing, see comment Figure 6.

We agree with this reviewer on the importance to clarify the question of tumor formation. We also appreciate that defining tumors as fast- and slow-growing tumors may be misleading. We speculate that tumors only differ in the initial conditions during cell injections (such as number of cells injected) and the actual rate of tumor growth may therefore be the same. For this reason, we adjusted the figure, rewrote this experimental part to be more accurate and understandable and added new data about Myo1C mutation

association with different pathways in human cancers. These findings are now part of figure 7.

We also performed the tumorigenic assays in mice with initial number of cells to be reduced to 1/3 and in this case no tumors were found in either genotype suggesting that initial number of cells does have an impact on the final size of the tumor. This data is now part of figure 6A.

Reviewer #2 (Remarks to the Author):

Tomas Venit and colleagues study the function of the nuclear myosin 1 (NM1) protein. NM1 is a chromatin remodeling protein. Using a CRISPR-Cas9-mediated knockout cell clone of mouse embryonic fibroblasts (KO-MEFs), the authors perform a long series of detailed and well-controlled experiments, focusing primarily on the function, homeostasis and gene-mediated integrity of mitochondria and their metabolism. Loss of NM1 from these normal MEFs leads to Warburg-like effect and oncogenic transformation of the fibroblasts in the absence of cooperating oncogenes or loss of tumor suppressor genes. This established NM1 as a tumor suppressor. These interesting results are presented in Figs 1-3 and. In Fig. 4 the paper tries to generate a mechanism of action of NM1 and presents data that potentially link NM1 to the oncogenic mTOR pathway. mTOR possibly phosphorylates NM1, and NM1 binds to genes that express transcription factors that regulate mitochondrial gene expression. NM1 is also shown to regulate mTOR signaling in a positive feedback mechanism. Finally, transcriptomic analyses of tumors generated by the NM1-KO MEFs are presented in Fig. 6 that present again misregulation of mitochondrial genes.

The paper presents a series of well-controlled experiments and combines high-throughput analyses to more focused analyses of specific molecular pathways. This reviewer has two major conceptual issues with the presented data: a) the potency of NM1 KO leading to tumorigenesis in mice is very high. One therefore would expect a similar function of NM1 in human tumors. Accordingly, the paper presents the data of Fig. 6L and 6M, which are very important but correlative. No evidence for an equivalent mechanism in human tumors is presented.

The nude mouse model which we used in the tumorigenesis experiments has mutation in the *Foxn1* gene and it is T cell-deficient, which makes the mice much more susceptible to cancers in comparison to the WT condition. Therefore, even though we believe NM1 depletion may also lead to tumorigenesis in human cancers, the incidence would be much lower in comparison to the immunosuppressed mouse model.

In the original version of the manuscript, *Myo1C* gene expression is heavily suppressed in several types of human cancers. To get more insights, during the revision of the manuscript we studied more closely the expression profile of all published cancer datasets and performed single sample gene set enrichment analysis followed by linear regression analysis. We found that proliferation, apoptosis and, most importantly, glycolysis pathways are upregulated in cancers with *Myo1C* mutations while in the same cancers the mTOR pathway is suppressed. This is in strong agreement with previous work from the lab (Venit et al., 2020) and supports our observations in the mouse model.

The new results together with previous results presented as Figure 6L and 6M are now put together in figure 7 of the revised manuscript.

b) It is unclear which tumor suppressor signaling pathway mediates NM1 phosphorylation via mTOR, which then maintains mitochondrial homeostasis by regulating transcription of PGC1a and TFAM. This core question remains unanswered and characterizes the essential message of this paper.

Thanks for raising this point but it remains a little unclear to us. Especially, we are not sure what is meant by “which tumor suppressor signaling pathway mediates NM1 phosphorylation via mTOR”.

Two main tumor suppressor pathways – p53 pathway and RB pathway – are not really associated with the mTOR pathway directly and, therefore, they don't have substantial effect on mTOR target genes via mTOR phosphorylation. Here, we show that Rapamycin which is a specific inhibitor of the mTORC1 complex reduces association of NM1 with the TSS of TFAM, PGC1a and mTOR itself. Since Rapamycin inhibits the kinase activity of mTORC1 taken altogether these results suggest that NM1 is either directly or indirectly phosphorylated by mTOR. We also showed that the upstream PI3K-AKT signaling is suppressed in NM1 depleted cells suggesting that there are some feedback mechanisms to regulate the whole PI3K-AKT-mTOR pathway. We agree with this reviewer that understanding how regulation of the whole cascade is performed is critical. However, dissecting the molecular mechanisms would go beyond the scope of the present study (how NM1 mediates transcriptional regulation of mitochondrial function thus affecting metabolic reprogramming) and it is part of a follow-up study. Further clarifications are now included in the revised version of the manuscript.

I here present some additional comments on specific results of this paper.
1. Title: “Regulation of OXPHOS...” is rather vague. It would be nice if the title indicated more accurately the proposed mechanism: positive regulation of genes that promote mitochondrial biogenesis.

As per reviewer's suggestion we have now changed the title to “Positive regulation of oxidative phosphorylation by nuclear myosin 1 protects cells from metabolic reprogramming and tumorigenesis in mice”

On the other hand, we have not included the term “mitochondrial biogenesis” as we only found mitochondrial quality control genes to be deregulated while mitochondrial fission and fusion or mitophagy do not seem to be affected. Also, even though we thoroughly phenotyped mitochondria in NM1 KO cells, our main focus was on energy metabolism and we haven't studied other mitochondrial functions.

2. The RNAseq data of Fig. 1E are difficult to understand. The legend indicates that the genes

are listed in descending order. What does this mean? Most genes show the same expression pattern in WT and a different pattern in KO MEFs. The genes at the bottom of the lists show the inverse pattern. This is not explained. Why is this for only a few genes in each of the 5 complexes?

Thank you and we apologize for the mistake in the manuscript which is now fixed. We have also improved the explanation of the data in the corresponding figure legend and main text. Heatmaps are based on the RNA seq data from primary mouse embryonic fibroblasts directly isolated from NM1 WT and KO embryos (Venit et al., 2020). We plotted all differentially expressed nuclear-encoded OXPHOS genes and organized them based on 1) the mitochondrial respiratory complex they belong to, and 2) their expression profiles. In each complex, genes are listed in ascending order based on log₂ fold change of their expression in KO in comparison to WT. Therefore, at the top of each column there are genes which are downregulated the most in the KO cells (strongest red in KO column), while at the bottom there are genes which are most upregulated in the KO cells (strongest blue in KO column). The conclusion of this figure is, that while there are a few OXPHOS genes upregulated in NM1 KO primary MEFs, the majority of genes are suppressed which correlates with our experimental data obtained from stable NM1 KO MEFs prepared by CRISPR/Cas9 technology (Figure 1A-1D).

3. The data of Suppl. Fig. 2C and 2D are important but there are not explained to a non-specialist. What is A and B and what do the changes supposedly represent? Based on these data and the ATACseq data the authors conclude in p.10, lines 292-294 that NM1 deletion does not affect the expression of mitochondrial genes directly by changing the chromatin landscape or global 3D genome architecture, ... through direct regulation of ...transcription factors. The terms directly and direct generate confusion. Which protein directly changes genome architecture? The histones? If NM1 acts as a chromatin remodeling factor, how are the ATACseq and HiC data interpreted? Is the impact of NM1 to the specific transcription factors of mitochondrial genes dependent on the ability of NM1 to regulate chromatin remodeling?

The eukaryotic genome is hierarchically organized in A and B compartments respectively characterized by open and compact chromatin. Therefore, A and B compartments match euchromatic and heterochromatic states compatible with active and repressed transcription, respectively. We recently reported that β -actin is a key regulator of genome organization at compartment level (Mahmood et al 2021, Nature Communications). We reported that depletion of the nuclear β -actin pool leads to compartment switching from compartment A to compartment B and, vice versa, from compartment B to compartment A. The primary consequence of compartment switching is that some of the genes originally localized in transcriptionally inactive compartments B become activated. The same applies to genes that in the wild type condition are in the transcriptionally active compartment A whereas upon nuclear β -actin depletion they become inactive due to a switch from compartment A to transcriptionally repressed compartment B. The take home message of the paper is that β -actin, as part of the chromatin remodeling complex SWI/SNF (BAF), is required for the functional architecture of genome.

Given the above results, in the present study, we tested if NM1 as an actin binding protein, plays a role in genome organization as well. Results from HiC seq experiments suggest that genomic architecture is not affected by the absence of NM1 as demonstrated by the lack of compartment switching and/or changes in TADs. So, taken altogether these results

suggest that NM1 does not regulate transcription by controlling the 3D organization of the genome but rather by acting at the gene level. Examples are genes such as TFAM, PGC1a or mTOR, as well as the p53 target Cdkn1a (p21) gene. In this case, at the gene promoter NM1 interacts with p53 to contribute to local rearrangement of chromatin for p21 transcription activation (Venit et al., 2020).

The mechanism seems to be conserved as NM1 regulates the epigenetic landscape required for transcription activation (see Percipalle et al 2006, Sarshad et al 2013, 2014, Almuzzaini et al 2015, Venit et al 2020). This is corroborated by the distribution of ATAC-seq and ChIP-seq signals for SNF2H, H3K9ac and H3K4me3 along TFAM, PGC1a and mTOR. We detected a reduction in ATAC-seq signal and a drop in the majority of epigenetic marks in NM1 KO cells. ChIP-seq profiles across TFAM, PGC1a and mTOR, supporting our ChIPqPCR data (see main text), are now added as supplementary figure 3 and the results are discussed in the main text. We also clarified the nomenclature and rewrote the part about the effect of NM1 on global changes in 3D genome architecture.

4. Related to the above comment, does the ChIP-qPCR analysis of Fig. 4 agree with the ATACseq analysis of the PGC1a, TFAM and mTOR genes?

See answer to the previous point. In brief, results from ATAC-seq data agree with those from ChIPqPCR experiments. The data are now part of the supplementary figure 3.

5. The data of Fig. 4G on NM1 phosphorylation are very interesting. Phospho-peptide motif analysis for mTOR targets is worth performing and verification of the phosphorylation by immunoprecipitation-western blot analysis.

Motif analysis on NM1 showed the presence of GSK3 β binding sites in the NM1 C-terminus. These predictions were corroborated by phosphoproteomic analysis of NM1 which identified a specific phosphorylation site on Serine position 1020 in the NM1 C-terminus. This residue is targeted by GSK3 β during G1 phase of the cell cycle, it is required for chromatin binding by NM1 at transcription start site and it protects NM1 from proteasomal degradation (Sarshad et al., 2014). Currently we don't know whether mTORC1 phosphorylation of NM1 is direct or indirect and whether it occurs on the same site and we are planning to investigate this specific aspect as part of a follow up study. However, we speculate that mTOR could regulate NM1 indirectly via GSK3 β -dependent phosphorylation as it has been suggested by several other studies (Zhang et al. 2006, Evangelisti et al. 2020, Bautista et al. 2018). We discuss this in more detail in the revised manuscript.

6. P.11, lines 327-329: NM1 functions as part of a nutrient-sensing signaling pathway. The evidence that supports this evidence should be clearly presented.

The term “nutrient-sensing signaling pathway” refers to the PI3K-AKT-mTOR signaling pathway as per several publications (see for instance Hong et al., 2011; Sabatini, 2017; Huynh et al., 2023; Tan et al., 2016). However, we agree with this reviewer that it is a bit misleading as there can be other nutrient sensing pathways other than mTOR which are

not studied in this paper. We have, therefore, replaced the terminology “nutrient-sensing signaling pathway” with “mTOR-signaling pathway” in the revised manuscript.

7. The potency of NM1 KO in generating tumors in mice requires explanation.

Thanks for pointing this out. We have, accordingly, adjusted the figures related to the tumorigenesis part, rewritten the results, added some supportive literature and discussed more the results in the revised text to make our statements clearer and more understandable.

8. As commented above, the paper contends that nutrient sensing is at play upstream of NM1 regulation. Yet Fig. 7 indicates growth factor signaling as acting upstream. Are growth factors implied by the term “nutrient-sensing”?

As mentioned above, the “nutrient-sensing” terminology is replaced with “mTOR-signaling pathway” across the entire manuscript.

Reviewer #3 (Remarks to the Author):

This is a potentially interesting study but can not be meaningfully evaluated because it is completely missing an absolutely critical control - stable reconstitution (addback) of NM1 in NM1-knockout cells. Creating addback controls is important for any KO cell line but is especially important in clonal KO lines which these appear to be based on the referenced previous publication from this group (Venit et al Commun Biol 2020). These control cells are required to show that the NM1-KO phenotypes can be rescued by adding back NM1 and that the differences between cell lines did not arise due to off-target effects of CRISPR, unrelated genetic or epigenetic changes that arose during KO cell line establishment, or other random reasons. Even better, if a loss of function point mutant of NM1 can be created that lacks activity, this should also be used to make an a control cell line in addition to WT NM1. Unfortunately, without this basic control it is pointless to evaluate the data in depth.

I would be happy to review an updated version of this manuscript that includes NM1 addback controls for most experiments.

We would like to thank this reviewer for the insightful comment and we agree that the off-targeting of CRISPR guiding RNA could cause a misinterpretation of the observed phenotype.

However, we would like to emphasize that, our guiding RNA molecules were prepared in a way that possible off-targets are localized in the noncoding part of the genome (see Venit et al 2020). Secondly, in the previous paper we introduced three model systems – 1. NM1 KO MEFs prepared by CRISPR/Cas9, 2. Primary mouse embryonic fibroblast isolated from NM1 KO mice which were prepared by Cre/LoxP homologous recombination and 3. NM1 knockdown by siRNA. All these cells phenocopy and complement each other in the observed phenotypes (Venit et al Commun Biol 2020). In accordance with that, in this paper we performed the RtgPCR and immunoblots of Oxidative phosphorylation genes in CRISPR KO cell lines and compared the data to OXPHOS gene expression in primary embryonic fibroblasts, again showing the same phenotype between the two experimental model systems (Figure 1). This in principle supports the specificity of our NM1 KO model system.

That said, we agree with this reviewer that rescue add-back experiments are very important controls. For this reason, we reintroduced exogenous HA-tagged NM1 in the NM1 KO MEFs. In this system HA-tagged NM1 is constitutively expressed. We used these cells to run some of the key experiments in the study. Namely, we performed RtgPCR of Oxidative phosphorylation genes, glycolytic genes, TFAM, PGC1a and mTOR. We also stained the cells with mitotracker dyes to evaluate mitochondrial content, we measured metabolites with the highest difference between WT and NM1 KO conditions. Finally, using the new cells constitutively expressing HA-tagged NM1, we performed ChIPqPCR analysis at TFAM, PGC1a and mTOR TSS. As can be seen in the revised manuscript, in all aforementioned experiments, expression of HA-tagged NM1 in the KO background rescued phenotypes and phenocopies observations in the WT cells. All data are now part of the supplementary figure 1.

We conclude that the NM1 KO phenotypes can be rescued by adding back NM1 and that the differences between cell lines is not a consequence of off-target effects of CRISPR,

unrelated genetic or epigenetic changes that arose during KO cell line establishment, or other random reasons.

REVIEWER COMMENTS

Reviewer #1 (Remarks to the Author):

Many of my initial question have been answered and clarified, even if several of the questions raised will be followed up in the next study.

The response from the authors to my and reviewer 2's concerns together with the re-writes taken place merits the manuscript publication.

Reviewer #3 (Remarks to the Author):

In my original comments my main point was that the study lacked NM1 addback controls. The conclusions of the study rely heavily on comparisons of WT vs KO cell lines which can spontaneously develop phenotypic differences when being derived. Demonstrating that NM1 addback to KO cells rescues the difference between WT and KO cells is not just important, its essential for demonstrating specific NM1-dependent effects. The authors have created an addback cell line but have not used it in a way that meaningfully addresses this criticism for two main reasons:

1) Some major claims (section titles) remain completely unsupported by experiments that include addback controls. These include "NM1 depletion results in dysregulated mitochondrial dynamics" and "NM1 KO cells exhibit a metabolome profile typical for cancer cells". Even if one phenotype is ultimately shown to be rescued that doesn't mean that another phenotype will be rescued and so addback controls are needed for each phenotype/experimental line.

2) Even lines of experiments that do include the new KO+NM1 addbacks only present comparisons of WT vs. KO+NM1 addback cells, which is not the most important comparison. I can't remember ever seeing a WT vs KO+addback comparison in a publication without the KO line also included in the comparison. The most relevant comparison is the KO(+empty vector) vs. KO+NM1 addback. Ideally a comparison of WT vs. KO vs. KO+NM1 addback would be shown but at the very least KO vs. KO+NM1 addback should be shown.

Furthermore, the add-back controls are not just nice to have, they are essential pieces of data and they should be shown in the main figure and not simply in the supplement.

The authors point out that they look at both CRISPR KO cells and primary KO MEFs in figure 1. This is a step in the right direction but is not a substitute for using addback controls and directly comparing KO vs KO+addbacks. In addition, the extent to which the expression phenotypes of the CRISPR KO and primary KO cells overlap is not as clear as it seems at first. The authors point out in the text that *Uqcrb* goes in opposite directions in the two sets of cells but the overlap in the two data sets is even thinner. Two out of four nuclear-encoded proteins shown in Fig. 1A are not listed in Fig. 1E. Furthermore, four out of six nuclear encoded transcripts shown in Fig. 1C are either not listed in Fig. 1E or don't change in the same direction as in Fig. 1E (*Atp5d* doesn't change in 1C but goes down in 1E in KO cells, while *Uqcrb* goes down in 1C but up in 1E in KOs). So really, expression levels of only four nuclear genes go in the same direction between the two sets of cell lines. Of course, you wouldn't expect expression levels to be completely the same in both sets of KOs but that's the point, and that's why KO vs KO+NM1 addbacks should be analyzed – to more clearly show which changes are dependent on NM1. The CRISPR KO lines could have also undergone more extensive expression profiling like the primary MEF KOs so that more overlap could be shown between the two sets of KO cells.

There are other improvements that could be made. For instance, if the authors want to claim that NM1 loss suppresses mTORC1 signaling then they should actually demonstrate effects on mTORC1 signaling (and not just transcript levels) by performing western blots with phospho-S6K, total S6K, phospho-4E-BP, total 4E-BP (classic mTORC1 substrates).

In summary, all major claims about distinct phenotypes should be supported by experiments that include NM1 addback control cells. Furthermore, all experiments with addback cells should include a comparison of WT vs. KO vs. KO+addback lines to clearly show in a single experiment that a phenotype is arising in the KO and being rescued in the addbacks. Only comparing WT to KO+addback cells within an experiment is NOT sufficient to demonstrate rescue.

Reviewer #4 (Remarks to the Author):

Reviewer #2 was mainly concerned by two aspects of the original manuscript from Venit, Percipalle and colleagues: a) the relevance of the oncosuppressive effects of NM1 for human tumour biology b) the limited understanding of the signaling mechanism underpinning the oncosuppressive activity of NM1 demonstrated in murine models.

I share Reviewer#2 concerns and I found that the revised manuscript does not provide convincing experimental evidence about the relevance of NM1 to human tumor biology.

The data obtained from publicly available datasets on gene expression profiles and genetic alterations presented in figure 7 of the revised manuscript do not support the overall hypothesis that *Myo1C* is selectively inactivated in human tumors. The results of Figure 7 are allowing the Authors to reach trivial conclusions about p53 and mTOR mutational status in cancer, but do not convincingly demonstrate a role for NM1 in human tumor (e.g. page 15 of the revised manuscript “While p53 shows a very high

mutagenic rate in the majority of cancers, Myo1C and mTOR show relatively low levels of mutagenesis in different cancer tissues”.

Regarding the metabolic and signaling mechanism underpinning the oncosuppressive action of NM1 in mice, the model proposed remain vague as originally pointed out by Rev#2 (specific point 1). Result sections titles such as “NM1 regulates mitochondria by regulating specific mitochondrial transcription factors (page 10)” show the lack of clarity on the mechanisms. Other section titles such as the one on page 13 “NM1 is a novel tumor suppressor” should be toned down and contextualized to the murine species.

Overall while the results of NM1 deletion in murine MEFs are compelling and strengthened by the control experiments requested by Reviewer#3, their relevance to human disease initiation and progression remains largely speculative. Deletion and genetic reconstitution performed in human cancer lines or in pre-malignant human cells and their effects on xenografts growth would have strengthened the hypothesis that NM1 could have an oncosuppressive role in human setting.

REVIEWER COMMENTS

Reviewer #1 (Remarks to the Author):

Reviewer #1 comment: Many of my initial question have been answered and clarified, even if several of the questions raised will be followed up in the next study.

The response from the authors to my and reviewer 2's concerns together with the re-writes taken place merits the manuscript publication.

Authors response: We would like to thank the reviewer for the time spent on our manuscript and we very much appreciate the raised questions during the first revision.

Reviewer #3 (Remarks to the Author):

Reviewer #3 comment: In my original comments my main point was that the study lacked NM1 addback controls. The conclusions of the study rely heavily on comparisons of WT vs KO cell lines which can spontaneously develop phenotypic differences when being derived. Demonstrating that NM1 addback to KO cells rescues the difference between WT and KO cells is not just important, its essential for demonstrating specific NM1-dependent effects. The authors have created an addback cell line but have not used it in a way that meaningfully addresses this criticism for two main reasons:

1) Some major claims (section titles) remain completely unsupported by experiments that include addback controls. These include "NM1 depletion results in dysregulated mitochondrial dynamics" and "NM1 KO cells exhibit a metabolome profile typical for cancer cells". Even if one phenotype is ultimately shown to be rescued that doesn't mean that another phenotype will be rescued and so addback controls are needed for each phenotype/experimental line.

2) Even lines of experiments that do include the new KO+NM1 addbacks only present comparisons of WT vs. KO+NM1 addback cells, which is not the most important comparison. I can't remember ever seeing a WT vs KO+addback comparison in a publication without the KO line also included in the comparison. The most relevant comparison is the KO(+empty vector) vs. KO+NM1 addback. Ideally a comparison of WT vs. KO vs. KO+NM1 addback would be shown but at the very least KO vs. KO+NM1 addback should be shown.

Furthermore, the add-back controls are not just nice to have, they are essential pieces of data and they should be shown in the main figure and not simply in the supplement.

Authors response: Thanks for the insightful comments. We have now repeated the whole study with all three cell lines. In the revised version of the manuscript all experiments have been conducted with WT, KO and KO+NM1 cells. As can be seen, generally, reintroduction of NM1 in the KO background leads to full or partial rescue of the phenotype.

Specifically, regarding the question by this reviewer on "NM1 depletion results in dysregulated mitochondrial dynamics", using WT and KO we show that while markers for mitochondrial fission (see unaltered levels of DNM1L and Fis1 gene expression in the three cell lines) are not affected, markers involved in other mitochondrial turnover steps are dysregulated, including mitochondrial fusion (increased Mfn1 levels), quality control (decreased expression of both Pink1 and alpha-Synuclein) and mitophagy (decreased Becn levels). Interestingly, NM1 expression in KO cells (KO+NM1) could not rescue expression of Mfn1 and Becn proteins, while expression of Pink1 and Snca are rescued (Figure 3A). In addition, fission proteins Fis and Dnm1L whose levels were not changed appeared to be overexpressed in KO+NM1 cells (Figure 3A).

Similarly, regarding the concern by this reviewer on "NM1 KO cells exhibit a metabolome profile typical for cancer cells", we performed a new metabolomic profiling experiment with all three cell lines. The results clearly show the highest difference between KO and KO+NM1 cells, followed by WT and KO cells and then KO+NM1 and WT cells, suggesting that metabolic profiles of cells are dependent on the amount of NM1 and that reintroduction of NM1 can rescue KO phenotypes (please see the plot of PC3 values in panel B, supplementary figure 2). Further in-depth analysis shows that the NM1 reintroduction (KO+NM1 cells) has an impact on the abundance of amino acid metabolism and TCA cycle metabolic intermediates. KO+NM1 cells show reduction in the levels of metabolites such as betaine, L-isoleucine and Enol-phenylpyruvate, implicated in amino acid metabolism, to levels that are comparable to that of WT cells relative to KO cells. Among the TCA cycle key intermediates, we highlight succinic acid semialdehyde (SSA) and L-malic acid. We demonstrate that KO+NM1 cells show a reduction in the levels of SSA, which are significantly elevated in the KO cells. Remarkably, SSA is accumulated when the oxidation of succinic semialdehyde to succinic acid is impaired, a key intermediate metabolite of the TCA cycle. KO+NM1 cells also showed rescue of the levels of malic acid, which is a strong indication that the TCA cycle is no longer impaired in the NM1-rescued cells (Supplementary Figure 2E) supporting our original metabolomic data.

Interestingly, NM1 addback control cell line (KO+NM1) displays higher NM1 expression levels in comparison to expression of endogenous NM1 in WT cells. This has given us the possibility not only to rescue the majority of the observed phenotypes but also speculate on NM1 dosage dependent changes in some of the phenotypes including increased expression of Oxphos genes or TFAM and an increase in mitochondrial network formation in comparison to WT cells (see Figure 2, panel E). These considerations are interesting per se because they further support a direct role for NM1 in mitochondrial function. They also open interesting avenues for future research on the exact timing and the amount of NM1 needed for proper regulation of cell metabolism. For example, KO+NM1 rescue cells shows higher variability in the level of metabolites between samples; they cannot fully rescue mitochondrial membrane potential even though extensive mitochondrial network and cannot fully suppress the tumor growth as seen in WT cells. However, the RNA seq analysis of tumors derived from KO and KO+NM1 cells shows rescue of oxidative phosphorylation as well as other previously described NM1 KO phenotypes.

Taken altogether, our experiments indicate that KO+NM1 cells are able to rescue the main mitochondrial phenotypes caused by NM1 depletion. Since in KO+NM1 cells mitochondria themselves may not be entirely functional, it is possible that NM1 overexpression has a negative effect on the state of steady cellular conditions causing gene expression changes between mitochondria- and nuclear-encoded mitochondrial genes. At this stage these are speculation that

warrant further investigations but we are thankful to this reviewer for the addback control suggestions.

In summary, we hope that now, we have addressed all major concerns from this reviewer.

Reviewer #3 comment: The authors point out that they look at both CRISPR KO cells and primary KO MEFs in figure 1. This is a step in the right direction but is not a substitute for using addback controls and directly comparing KO vs KO+addbacks. In addition, the extent to which the expression phenotypes of the CRISPR KO and primary KO cells overlap is not as clear as it seems at first. The authors point out in the text that *Uqcrb* goes in opposite directions in the two sets of cells but the overlap in the two data sets is even thinner.

Authors response: We agree that the use of CRISPR KO and primary KO MEFs is important but the use of an addback control is even more important as pointed out by this reviewer. As mentioned above, all experiments were performed from scratch using the three cell lines WT, KO and KO+NM1.

Reviewer #3 comment: Two out of four nuclear-encoded proteins shown in Fig. 1A are not listed in Fig. 1E. Furthermore, four out of six nuclear encoded transcripts shown in Fig. 1C are either not listed in Fig. 1E or don't change in the same direction as in Fig. 1E (*Atp5d* doesn't change in 1C but goes down in 1E in KO cells, while *Uqcrb* goes down in 1C but up in 1E in KOs). So really, expression levels of only four nuclear genes go in the same direction between the two sets of cell lines. Of course, you wouldn't expect expression levels to be completely the same in both sets of KOs but that's the point, and that's why KO vs KO+NM1 addbacks should be analyzed – to more clearly show which changes are dependent on NM1. The CRISPR KO lines could have also undergone more extensive expression profiling like the primary MEF KOs so that more overlap could be shown between the two sets of KO cells.

Authors response: Indeed, we agree with this reviewer that one would not expect expression levels to be completely the same in both sets of KOs (CRISPR KO and primary KO MEFs). Thanks for this, figure 1 has now been revised accordingly and as mentioned above, addback controls are included in all experiments (WT, KO and KO+NM1)

Reviewer #3 comment: There are other improvements that could be made. For instance, if the authors want to claim that NM1 loss suppresses mTORC1 signaling then they should actually demonstrate effects on mTORC1 signaling (and not just transcript levels) by performing western blots with phospho-S6K, total S6K, phospho-4E-BP, total 4E-BP (classic mTORC1 substrates).

Authors response: We are very interested in further dissecting the synergy between NM1 and mTORC1. We are working on an independent study to address this specific point, but understanding the relationship between posttranslational regulation capacity of mTORC1 and transcriptional regulation of NM1 is beyond the scope of the investigation about the role of NM1 in the mitochondrial biogenesis. For example, 4EBP1 protein which needs to be phosphorylated by mTORC1 to be released from eukaryotic translation initiation factor 4E-complex to stimulate protein synthesis is completely lost in NM1 KO and fully restored in KO+NM1 cells (see figure below). We don't know whether the transcriptional loss of 4EBP1 is directly regulated by NM1 or its loss is only a consequence of some other signaling pathway to secure protein synthesis without

active mTORC1. Therefore, further research as part of independent studies needs to be performed to address the relationships between different pro-proliferative pathways in NM1 KO background

Reviewer #3 comment: In summary, all major claims about distinct phenotypes should be supported by experiments that include NM1 addback control cells. Furthermore, all experiments with addback cells should include a comparison of WT vs. KO vs. KO+addback lines to clearly show in a single experiment that a phenotype is arising in the KO and being rescued in the addbacks. Only comparing WT to KO+addback cells within an experiment is NOT sufficient to demonstrate rescue.

Authors response: We are grateful to this reviewer since taken altogether, the results obtained from running in parallel the three cell lines (WT, KO and KO+NM1) support a direct role of NM1 in mitochondrial function and fully address this reviewer's original concerns

Reviewer #4 (Remarks to the Author):

Reviewer #4 comment: Reviewer #2 was mainly concerned by two aspects of the original manuscript from Venit, Percipalle and colleagues: a) the relevance of the oncosuppressive effects of NM1 for human tumour biology b) the limited understanding of the signaling mechanism underpinning the oncosuppressive activity of NM1 demonstrated in murine models. I share Reviewer#2 concerns and I found that the revised manuscript does not provide convincing experimental evidence about the relevance of NM1 to human tumor biology. The data obtained from publicly available datasets on gene expression profiles and genetic alterations presented in figure 7 of the revised manuscript do not support the overall hypothesis that Myo1C is selectively inactivated in human tumors. The results of Figure 7 are allowing the Authors to reach trivial conclusions about p53 and mTOR mutational status in cancer, but do not convincingly demonstrate a role for NM1 in human tumor (e.g. page 15 of the revised manuscript "While p53 shows a very high mutagenic rate in the majority of cancers, Myo1C and mTOR show relatively low levels of mutagenesis in different cancer tissues").

Authors response: We would like to thank reviewer #4 for the comments on the revised manuscript but it seems there is a misunderstanding, perhaps due to the fact that this reviewer has gone through the original submission and not revision 1. We will try to clarify all the concerns.

Firstly, we have not stated anywhere in the text that "Myo1C is selectively inactivated in human tumors". Actually, the sentence used in the text and mentioned by the reviewer: "While p53 shows a very high mutagenic rate in the majority of cancers, Myo1C and mTOR show relatively low levels of mutagenesis in different cancer tissues" is used to show that Myo1C (NM1) expression in many

cancers is preserved and is not selectively inactivated. This is followed by the next figure saying that if Myo1C is regulated in some cancer, it is usually downregulated in comparison to mTor which is usually upregulated in cancers which, by definition, means that NM1 is a potential tumor suppressor while mTor is an oncogene. However, we agree with both reviewers that providing only this data was entirely correlative. Therefore, we performed a single sample gene set enrichment analysis followed by linear regression analysis, to find the pathways which are associated with the Myo1C mutations in human cancers and found Glycolysis to be significantly enriched and mTor pathway to be significantly suppressed in Myo1C mutated cancers in comparison to cancers without Myo1C mutation, which is in agreement with the data observed in the NM1 KO mouse model. These results are presented in figure 7C and 7D.

Secondly, the whole study is performed on a murine model and we do not make any strong statements about human cancers. In fact, even the title of the article has a statement that deletion of NM1 leads to tumorigenesis in mice. The link to human cancers is only mentioned in the last part of results where we study and suggest the possibility of similar NM1-dependent effects on human cancers as part of figure 7. For further clarification, in the revised manuscript we changed the sentences mentioning the human tumorigenesis so they reflect our results.

See below for an example:

“We conclude that in mouse model system, NM1 serves a role as a tumor suppressor and its deletion leads to changes in cell metabolism that are directly connected to tumorigenesis, which could have similar effects also in human tumors”

Reviewer #4 comment: Regarding the metabolic and signaling mechanism underpinning the oncosuppressive action of NM1 in mice, the model proposed remain vague as originally pointed out by Rev#2 (specific point 1). Result sections titles such as “NM1 regulates mitochondria by regulating specific mitochondrial transcription factors (page 10)” show the lack of clarity on the mechanisms. Other section titles such as the one on page 13 “NM1 is a novel tumor suppressor” should be toned down and contextualized to the murine species.

Authors response: We have modified the titles to specifically reflect the mechanism. In the revised manuscript the titles read now “NM1 regulates mitochondria by regulating mitochondrial transcription factors TFAM and PGC1 α ” and “NM1 deletion leads to tumorigenesis in mice”.

We are confused about the question raised by this reviewer on the lack of mechanism and we do not agree with this concern. In fact, in this paper using chromatin immunoprecipitation and deep sequencing in combination with RNA sequencing, we show a lot of evidence that NM1 binds to the transcription start sites of mTor, TFAM and PGC1 α regulating chromatin by facilitating recruitment of histone modifying enzymes. As one would expect for this type of mechanism, upon deletion of NM1, there is dysregulation of the epigenetic marks and a reduction in expression of these genes which leads to dysregulation of mitochondria, suppression of Oxidative phosphorylation and metabolic switch to aerobic glycolysis. Direct evidence for these mechanisms is now supported by our addback control experiment where we clearly show that reintroduction of NM1 in the KO background rescues the active chromatin landscape and transcription of the above genes. We believe that by redoing the whole study with the new control, we further strengthen our claims about the role of NM1 in regulation of cell metabolism.

Reviewer #4 comment: Overall while the results of NM1 deletion in murine MEFs are compelling and strengthened by the control experiments requested by Reviewer#3, their relevance to human disease initiation and progression remains largely speculative. Deletion and genetic reconstitution performed in human cancer lines or in pre-malignant human cells and their effects on xenografts growth would have strengthened the hypothesis that NM1 could have an oncosuppressive role in human setting.

Authors response: Thanks for these general comments. We agree that these results show compelling evidence that NM1 is required for transcriptional regulation of metabolic reprogramming in mice and lack of NM1 leads to tumorigenesis in mice. Indeed, this initial study focuses on the mouse as a model system to study metabolic reprogramming. However, the data in figure 7 derived from human cancers are used to point out the possibility that similar mechanisms may be happening in humans and this grants further investigations as part of independent studies.

REVIEWERS' COMMENTS

Reviewer #3 (Remarks to the Author):

1) The authors have addressed my major concern by including addback controls and presenting comparisons between NM1 WT, KO, and KO+NM1 addback for most experiments. I commend the authors for redoing those experiments.

The last three comments are related to mTORC1:

2) Regarding the involvement of mTORC1, the statement in the abstract/summary that “NM1 depletion leads to suppression of the PI3K/Akt/mTOR pathway” is not clearly supported by the existing data and making this claim in the abstract is not warranted. Yes, some pathway components are expressed at lower levels in NM1 KO cells and this is rescued in the KO+NM1 addbacks (Fig 4J). However, the effects on suppressors of the pathway, PTEN and TSC2, contradict the suggestion that NM1 activates mTORC1 signaling. PTEN levels are increased by NM1 addback while TSC2 levels go down in NM1 KO cells and are rescued by KO+NM1. These effects on PTEN and TSC2 would instead be consistent with NM1 acting to suppress mTORC1 signaling (or NM1 depletion activating mTORC1). Furthermore, since the authors do not show any signaling blots in the manuscript (as I suggested in my last round of comments), it is not possible to draw strong conclusions about the effects of NM1 on mTORC1 signaling. Indeed, the one 4EBP blot shown in the rebuttal is confusing, as the authors point out, since NM1 depletion appears to result in loss of 4EBP expression which would in effect be equal to activation of mTORC1 signaling (not suppression).

Therefore, the authors should remove any strong statements in the abstract and main text about the directionality and potency of effects of NM1 on mTORC1 signaling (perhaps in favor of some general statement in the conclusion about a possible reciprocal regulatory relationship that needs to be further explored).

3) The authors prefer to not get into the detailed mechanism of how mTORC1 controls NME1 in this manuscript (as they are working on this as a follow up project). This seems completely fine to me. They also prefer not to show any signaling blots. In this context, I suggest they remove the phos-tag blot in Fig. 4I from the manuscript completely. It really doesn't add anything since the effect/function of the phosphorylation event is not demonstrated. Speculating about the role of this site based on a small amount of data may be misleading (and perhaps only serves to scoop the next project a bit).

4) Related to Comment 3, I suggest the working model (Fig. 8) is adjusted to show a dotted line going from mTORC1 to NME1 perhaps with a question mark nearby. This is to make clear that the mechanism is completely unknown. Also, a question mark next to the phosphorylation site on NME1 would be best since the function of this site is not known (if that data was left in). Visually, the working model is very mTORC1-centric when in fact much of the paper is not really about mTORC1 regulating NME1.

Good luck!

REVIEWERS' COMMENTS

Reviewer #3 (Remarks to the Author):

Reviewer. The authors have addressed my major concern by including addback controls and presenting comparisons between NM1 WT, KO, and KO+NM1 addback for most experiments. I commend the authors for redoing those experiments.

Reply. We are thankful to this reviewer for the insights provided to improve our paper and we are happy to hear that we were able to address all comments.

Reviewer. The last three comments are related to mTORC1:

Regarding the involvement of mTORC1, the statement in the abstract/summary that “NM1 depletion leads to suppression of the PI3K/Akt/mTOR pathway” is not clearly supported by the existing data and making this claim in the abstract is not warranted. Yes, some pathway components are expressed at lower levels in NM1 KO cells and this is rescued in the KO+NM1 addbacks (Fig 4J). However, the effects on suppressors of the pathway, PTEN and TSC2, contradict the suggestion that NM1 activates mTORC1 signaling. PTEN levels are increased by NM1 addback while TSC2 levels go down in NM1 KO cells and are rescued by KO+NM1. These effects on PTEN and TSC2 would instead be consistent with NM1 acting to suppress mTORC1 signaling (or NM1 depletion activating mTORC1). Furthermore, since the authors do not show any signaling blots in the manuscript (as I suggested in my last round of comments), it is not possible to draw strong conclusions about the effects of NM1 on mTORC1 signaling. Indeed, the one 4EBP blot shown in the rebuttal is confusing, as the authors point out, since NM1 depletion appears to result in loss of 4EBP expression which would in effect be equal to activation of mTORC1 signaling (not suppression).

Therefore, the authors should remove any strong statements in the abstract and main text about the directionality and potency of effects of NM1 on mTORC1 signaling (perhaps in favor of some general statement in the conclusion about a possible reciprocal regulatory relationship that needs to be further explored).

Reply. As requested, we changed the text of the abstract to reflect more the presented data.

Reviewer. The authors prefer to not get into the detailed mechanism of how mTORC1 controls NME1 in this manuscript (as they are working on this as a follow up project). This seems completely fine to me. They also prefer not to show any signaling blots. In this context, I suggest they remove the phos-tag blot in Fig. 4I from the manuscript completely. It really doesn't add anything since the effect/function of the phosphorylation event is not demonstrated. Speculating about the role of this site based on a small amount of data may be misleading (and perhaps only serves to scoop the next project a bit).

Reply. We agree with this point and we have removed the mentioned figure from the text and it will be used in future studies. While, the effect/function of the phosphorylation event is not demonstrated, it is nevertheless important to underline that NM1 is directly phosphorylated by GSK3beta (see earlier publication from our lab, Sarshad et al 2014, PLOS Genetics) and this phosphorylation event is essential for chromatin binding at TSS and NM1 stability.

Reviewer. Related to Comment 3, I suggest the working model (Fig. 8) is adjusted to show a dotted line going from mTORC1 to NME1 perhaps with a question mark nearby. This is to make clear that the mechanism is completely unknown. Also, a question mark next to the phosphorylation site on NME1 would be best since the function of this site is not known (if that data was left in). Visually, the working model is very mTORC1-centric when in fact much of the paper is not really about mTORC1 regulating NME1.

Reply. We agree with these suggestions and we have made changes in the final scheme which is now more NM1-centered rather than mTOR-centered.